# Learning Explicit Circuit Representations for Quantum States from Local Measurements

## Abstract

Characterizing quantum states is essential for advancing many quantum technologies. Recently, deep neural networks have been applied to learn quantum states by generating implicit representations that map them into classical vectors. Despite their success in predicting state properties, these representations remain a black box, lacking insights into strategies for experimental reconstruction. In this work, we aim to open this black box by developing explicit representations of quantum states through the generation of preparation circuits using a reinforcement learning agent with a local fidelity reward function. Relying solely on measurement data from a few neighboring qubits, our agent accurately recovers properties of target states. Specifically, we design a quantum measurement feature aggregation block which is used to extract global features of quantum states from local measurement data. We also provide a theoretical guarantee for the proposed local fidelity reward function. Extensive experiments demonstrate the effectiveness of our framework in learning various quantum states of up to 100 qubits, including those generated by Instantaneous Quantum Polynomial circuits, evolved by Ising Hamiltonians, and many-body ground states. The learned circuit representations can be further applied to Hamiltonian learning as a downstream task utilizing a simple linear model.

## 1 Introduction

Quantum state characterization is a critical task in quantum information, underpinning the development of quantum computing, quantum communication, and quantum sensing technologies. There are two main approaches to tackling this task: classical methods and quantum methods. Classical methods, such as quantum state tomography (Tóth et al., 2010; Gross et al., 2010; Cramer et al., 2010; Lanyon et al., 2017; Cotler & Wilczek, 2020), reconstructs the quantum state by measuring an informationally complete set of observables. These methods require exponentially increasing sample complexity in measurements as the size of the quantum system grows, making them impractical for systems with many qubits and thereby limiting their applicability for practical use. Quantum methods, represented by variational quantum algorithms (Cerezo et al., 2021), utilize the power of quantum circuits to learn quantum states. These methods (Peruzzo et al., 2014; Farhi et al., 2014; Du et al., 2022; Wu et al., 2023a) typically optimize a parameterized quantum circuit to approach the target quantum state. Nevertheless, due to the necessity of calculating gradients with respect to circuit parameters, where the loss landscape is often highly flat, these methods often struggle with issues such as barren plateaus (McClean et al., 2018; Cerezo et al., 2021) and local minima (Anschuetz & Kiani, 2022; Huang et al., 2024), consequently affecting their performance in learning large-scale quantum systems.

To address these issues, recent approaches integrate machine learning techniques to characterize quantum systems. These methods have shown success in quantum state learning (Carleo & Troyer, 2017; Sharir et al., 2020; Zhu et al., 2022; Zhang & Di Ventra, 2023; Tang et al., 2024a; Chen & Heyl, 2024; Du et al., 2023; Qian et al., 2024), quantum process learning (Huang et al., 2023; Torlai et al., 2023; Zhu et al., 2023), quantum property estimation (Zhang & Di Ventra, 2023; Wu et al., 2023c; Lewis et al., 2024; Tang et al., 2024a), quantum state classification (Tang et al., 2024b), quantum sensing (Xiao et al., 2022; Zhou et al., 2023) and quantum verification (Wu et al., 2023b; Qian et al., 2024). Through leveraging neural networks to learn efficient representations of

Table 1: Summary of quantum state characterization methods. #Observables: The number of observables utilized for characterizing the target quantum states. Experimental reconstructability: The ability to construct a quantum circuit that reproduces the state from its measurement data. Downstream applicability: The capacity to perform downstream tasks, such as Hamiltonian learning, based on the classical representation of the state. Scalablity: The ability to extend the learning scheme to large-scale quantum systems (e.g., $N > 20$ qubits).

| | Methods | #Observables | Experimental reconstructability | Downstream applicability | Mitigate Barren plateaus | Scalability |
|---|---|---|---|---|---|---|
| Quantum | Peruzzo et al. (2014) | - | ✓ | ✗ | ✗ | ✗ |
| | Farhi et al. (2014) | - | ✓ | ✗ | ✗ | ✗ |
| Classical | Tóth et al. (2010); Cotler & Wilczek (2020) | $\mathcal{O}(2^N)$ | ✗ | ✗ | - | ✗ |
| | Carleo & Troyer (2017); Chen & Heyl (2024) | $\mathcal{O}(2^N)$ | ✗ | ✓ | - | ✗ |
| | Zhu et al. (2022) | $\mathcal{O}(N)$ | ✗ | ✓ | - | ✓ |
| | Ours | $\mathcal{O}(N)$ | ✓ | ✓ | ✓ | ✓ |

quantum states, low dimensional vectors, these methods significantly reduce the number of measurements required. Methods such as generative neural networks (GQNQ) (Zhu et al., 2022) and LLM4QPE (Tang et al., 2024a) aim to approximate the quantum state or its properties with fewer measurements by exploiting the underlying patterns and correlations present within a family of quantum states. By learning compact and expressive representations, these machine learning-based techniques offer scalable solutions for quantum state characterization, making them particularly valuable for learning large, complex quantum systems where traditional methods are infeasible. However, despite their advantages, these representations are often implicit. While they capture essential features and properties of the quantum state, they do not allow for the direct reconstruction of the state from the representation itself. This limitation poses challenges in scenarios where an explicit reconstruction of the quantum state is necessary, e.g., quantum phase estimation (Kitaev, 1995) and quantum simulation (Georgescu et al., 2014).

In this work, we propose a novel type of explicit circuit representations to characterize quantum states and design a deep reinforcement learning-based framework named QCrep to learn such representations that can experimentally reconstruct the target states. The circuit representation is a sequence of classical descriptions of the quantum circuit used to prepare the target state. This representation features the scalability and downstream task applicability of machine learning-based representations while achieving the experimental reconstructability. A comparison of different methods for state characterization is shown in Table 1. Two main challenges for learning the circuit representation are the high measurement overhead and the barren plateaus problem. The high measurement overhead roots from the fact that exponential number of measurements is required to fully characterize an unknown quantum state. However, for many practical cases, only specific properties of the states are of interests, making it unnecessary to reconstruct the full state. Therefore, we only use local measurements on a few neighboring sites of the quantum states to construct a local state representation for the target state. Additionally, we propose a novel Transformer-based (Vaswani et al., 2017) measurement feature aggregation block to recover properties of target states from local measurement data. To mitigate the problem of barren plateaus and local minima, we involve deep reinforcement learning that does not require computing gradients with respect to the circuit parameters. Besides, we design a novel reward function based on local fidelity, and provide a theoretical guarantee for the effectiveness of reconstructing global properties given local fidelity information of the states. The contributions are:

(1) We develop a novel type of representations for quantum states, termed the explicit circuit representations. Unlike conventional implicit state representations in GQNQ (Zhu et al., 2022) and Neural Quantum State (NQS) (Carleo & Troyer, 2017; Sharir et al., 2020; Zhang & Di Ventra, 2023; Chen & Heyl, 2024), circuit representations can be directly utilized to experimentally reconstruct the target states locally, which allows for computing the properties of interest via measuring the output states. Moreover, they possess the advantage of implicit representations that can be applied to downstream tasks.

(2) We design a reinforcement learning-based framework named QCrep to learn the explicit circuit representations for specific families of quantum states using only measurement data from a small number of neighboring sites. The circuits learned by this framework can construct quantum states with high global fidelity to the target states, utilizing $\mathcal{O}(N)$ number of observables with respect to

system size $N$. Benefiting from reinforcement learning with our novel reward function based on local fidelity, our framework circumvents the need for gradient-based optimization of circuit parameters and mitigates the barren plateaus problem, enabling scalability to larger systems. Notably, our framework is capable of reconstructing quantum states with up to 100 qubits.

(3) We experimentally demonstrate the effectiveness of our framework by learning four different families of target states and applying it to Hamiltonian learning (Wiebe et al., 2014; Wang et al., 2017) as a downstream task. Our framework shows superior performance in learning states generated by Instantaneous Quantum Polynomial (IQP) circuits (Bremner et al., 2010), states evolved by Ising Hamiltonians, and ground states of many-body quantum systems. For the downstream application, numerical experiments reveal that the unknown parameters of Hamiltonians can be accurately learned from local measurement data of their corresponding ground states, leveraging only a linear model acting on the circuit representations. This further highlights the versatility and effectiveness of our framework.

## 2 LEARNING EXPLICIT CIRCUIT REPRESENTATIONS FOR QUANTUM STATES

### 2.1 TASK DEFINITION

We define the task of learning explicit circuit representations for quantum states as characterizing a family of unknown quantum states $\mathcal{S} = \{\rho_s\}_s$ by constructing quantum circuits $\mathcal{U} = \{U_s\}_s$ that can prepare these states with high local fidelity, so that the reconstructed states can be directly measured to predict quantum properties of interests. We assume that the states can only be accessed in a black-box manner, meaning one can measure the states using measurement operators $\mathcal{M}$ but remains agnostic to the underlying circuits used for their preparation. Additionally, we assume that the measurement operators can only act on neighboring sites of the quantum states, a setup we refer to as local measurements. This measurement configuration has been widely adopted in prior works on quantum state characterization (Lanyon et al., 2017; Friis et al., 2018; Zhu et al., 2022; Kurmapu et al., 2023; Guo & Yang, 2023; Wu et al., 2023c) due to its feasibility for experimental realization. For this learning task, we do not put explicit constraints on the global fidelity between the reconstructed states and the target states, but focus on maximizing the average local fidelity.

**Explicit circuit representations.** Let $\rho_s$ be an $N$-qubit quantum state and $U_s$ the quantum circuit used to prepare it. $U_s$ can be expressed as a product of unitary gates, i.e., $U_s = \prod_t U_{s,t}(\phi_{s,t})$, where $U_{s,t}$ represents the quantum gates applied at time step $t$, and $\phi_{s,t}$ denotes the corresponding parameter(s) for those gates. The reconstructed state $\rho_s = U_s|0\rangle\langle0|^{\otimes N}U_s^\dagger$ has high average local fidelity with the target state $\rho_s$. The explicit circuit representation of $\rho_s$ is a sequence of $(u_{s,t}, \phi_{s,t})_t$, where $u_{s,t}$ is the classical description of the gate type of $U_{s,t}$.

**Overview.** To learn the circuit representations, we first perform local measurements on the target states, which is introduced in Section 2.2. After that, we design a reinforcement learning-based framework, QCrep, to decode the measurement data into quantum circuits, and keep the classical descriptions of the circuits as the circuit representations. This is described in Section 2.3, wherein a measurement feature aggregation block is proposed to process the local measurements, and a local fidelity reward function is designed to ensure learnability. Background information on quantum computation is introduced in Appendix B.

### 2.2 MEASUREMENT SETUP

We consider a set of measurements $\mathcal{M} = \{\boldsymbol{M}_i\}_{i=0}^{N-2}$, termed local measurements, performed on neighboring sites of the unknown $N$-qubit quantum states $\rho_s$. Each measurement $\boldsymbol{M}_i = (M_{ij})_{j=1}^K$ is a positive operator-valued measure (POVM) acting on two neighboring qubits $(i, i+1)$ of $\rho_s$, satisfying $\sum_{j=1}^K M_{ij} = I$. Specifically, we select the measurement operators $M_{ij}$ as the tensor product of two single-qubit Pauli operators, i.e., $M_{ij} \in \{X, Y, Z\}^{\otimes 2}$. We measure each pair of neighboring qubits using all such operators in a fixed order, taking the expectation values of the measurements to obtain the measurement output $\boldsymbol{m}_i \in \mathbb{R}^K$, where $K = 9$. We repeat this process for all qubit pairs and record the measurement data as $\boldsymbol{m} \in \mathbb{R}^{(N-1)\times K}$.

Importantly, the measurement operators are discarded when we input the measurement data into the agent. The correspondence between the operators and their expectation values is expected to be reconstructed during training. It is noteworthy that although quantum states and measurement operators are represented by complex-valued numbers, the measurement expectation values are real and range from $-1$ to $1$, since the eigenvalues of Pauli operators are either $-1$ or $1$. This property, along with the removal of measurement operators from the neural network's input, exempts the neural network from the overhead of processing complex values.

## 2.3 QCREP FRAMEWORK

To construct circuit representations for reproducing a family of quantum states, we design a reinforcement learning-based framework, QCrep. This framework relies exclusively on local measurements and avoids performing gradient descent on circuit parameters, effectively mitigating the barren plateau problem. The overall pipeline is shown in Figure 1. A deep reinforcement learning agent utilizing a neural network policy is employed to construct the circuit representations for a family of unknown quantum states $\mathcal{S}$. The environment in which the agent interacts and learns is defined as the quantum system. This environment is initialized with the quantum state to be learned, $\rho_s^{(0)} = \rho_s \in \mathcal{S}$, and is responsible for applying gates to the state as the agent iteratively learns to reconstruct the state. The observations are the local measurement values $\boldsymbol{m}_s$. We define the actions that the agent can take at step $t$ as applying a layer of quantum gates to the quantum state. The reward function is the local fidelity reward defined in Equation 4.

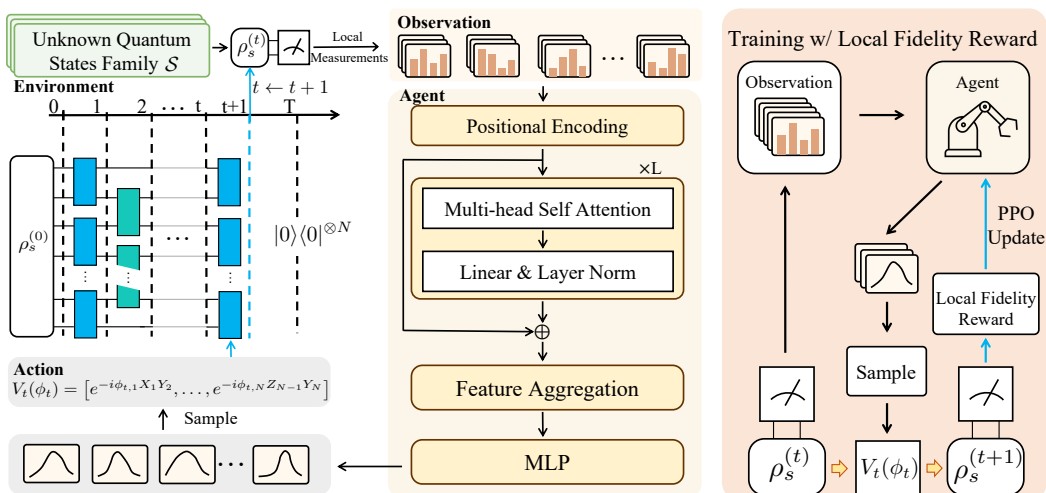

Figure 1: **QCrep framework.** Given an initial state $\rho_s^{(0)}$ sampled from an unknown quantum states family $\mathcal{S}$, the agent iteratively applies quantum gates $V_t(\phi_t)$ to evolve the state towards $|0\rangle\langle 0|^{\otimes N}$. The policy is parameterized by a neural network, which includes an Attention-based measurement feature aggregation block followed by a Multilayer Perceptron (MLP). The agent is trained using the PPO algorithm with a local fidelity reward.

Instead of directly learning $U_s$, the agent is trained to construct $V_s = \prod_{t=1}^{T} V_{s,t}(\phi_{s,t}) = U_s^{\dagger}$, which evolves $\rho_s$ towards $|0\rangle\langle 0|^{\otimes N}$, where $V_{s,t}$ represents a layer of quantum gates chosen at step $t$, and $\phi_{s,t}$ is the corresponding gate parameter. This approach enables the learning of a family of quantum states, as directly learning $U_s$ requires a fixed input state $|0\rangle\langle 0|^{\otimes N}$, which limits it to learning a single state. In contrast, by evolving towards $|0\rangle\langle 0|^{\otimes N}$, any state can be set as the input, facilitating the learning of a family of states. The target $U_s$ can then be obtained via taking the inverse of $V_s$, i.e., $U_s = V_s^{\dagger} = \prod_{t=T}^{1} V_{s,t}^{\dagger}(\phi_{s,t})$.

The entire process of learning the circuit representation for $\rho_s$ consists of several iterative steps. At each step $t$, the state $\rho_s^{(t)}$ is measured using local measurement operators and the agent takes the expectation values $\boldsymbol{m}_s^{(t)}$ as observations from the environment. The agent selects the action

$V_{s,t}(\phi_{s,t})$ according to its policy $\pi_{\boldsymbol{\alpha}}$, which is parameterized by a trainable Gaussian distribution generated from a neural network composed of a feature aggregation block followed by a Multilayer Perceptron (MLP). The action $V_{s,t}(\phi_{s,t}) = \bigotimes_k V_{s,t,i}(\phi_{s,t,i})$ is a column of single-qubit or two-qubit gates acting in parallel to every qubit $i$, where $V_{s,t,i}(\phi_{s,t,i}) = \exp(-i\phi_{s,t,i}G)$ are generated from the linear combination of the single-qubit and two-qubit Pauli operators

$$G \in \text{span}\left(\{X,Y,Z\} \cup \{X,Y,Z\}^{\otimes 2}\right). \tag{1}$$

To further reduce the search space, we apply a task-aware fashion to select a subset of gates as the action space, which will be described in detail in Section 3. After that, the environment updates the quantum state as $\rho_s^{(t+1)} = V_{s,t}(\phi_{s,t})\rho_s^{(t)}V_{s,t}^\dagger(\phi_{s,t})$ and the agent receives a reward $r^{(t)}$ defined in Equation 7. We repeat the above procedure until the average local fidelity $L(\rho_s^{(t)}, |0\rangle\langle 0|^{\otimes N})$, defined in Equation 4, exceeds a threshold of $1-\epsilon$, or until the number of iterative steps reaches a predefined maximum of $T$. We set $\epsilon = 0.001$ in the experiments. Note that this $T$ can be flexibly adjusted to control the accuracy of the reconstructed states or to meet hardware requirements when implemented on real quantum computers. The measurement complexity scales linearly with $T$, because for each $t$, only constant number of measurements is performed if the system size is fixed. The policy $\pi_{\boldsymbol{\alpha}}$ is updated using Proximal Policy Optimization (PPO) algorithm (Schulman et al., 2017),

$$\boldsymbol{\alpha}_{k+1} = \arg\max_{\boldsymbol{\alpha}} \mathbb{E}_{(\boldsymbol{m}, V(\phi)) \sim \pi_{\boldsymbol{\alpha}_k}}[J(\boldsymbol{\alpha}, \boldsymbol{m}, V(\phi), \boldsymbol{\alpha_k})], \tag{2}$$

and

$$J(\boldsymbol{\alpha}, \boldsymbol{m}, V(\phi), \boldsymbol{\alpha_k}) = \min\left(\frac{\pi_{\boldsymbol{\alpha}}(V(\phi)|\boldsymbol{m})}{\pi_{\boldsymbol{\alpha}_k}(V(\phi)|\boldsymbol{m})} A^{\pi_{\boldsymbol{\alpha}_k}}, \text{clip}_\delta\left(\frac{\pi_{\boldsymbol{\alpha}}(V(\phi)|\boldsymbol{m})}{\pi_{\boldsymbol{\alpha}_k}(V(\phi)|\boldsymbol{m})}\right) A^{\pi_{\boldsymbol{\alpha}_k}}\right), \tag{3}$$

where $A^{\pi_{\boldsymbol{\alpha}_k}}$ is the estimated advantage function associated with reward $r$, and $\delta$ measures the gap between the new and old policies. Finally, we keep the sequence of classical descriptions of the quantum gates $(v_{s,t}^\dagger, \phi_{s,t})_{t=T}^1$ as the circuit representation of $\rho_s$.

**Attention-based Measurement Feature Aggregation Block.** We construct a novel feature aggregation block to map the quantum measurement data $\boldsymbol{m}$ to a compact vector representation $\boldsymbol{p}$. There are two main features for this block: (1) A Transformer (Vaswani et al., 2017) module is proposed to capture the entanglement property of the quantum states from local measurement data. Due to the entangled nature of quantum states, non-local correlations exist among qubits, leading to long-range dependencies between measurement values. Therefore, we utilize self-attention to model the dependencies between different qubits. (2) An aggregation layer, implemented as global average pooling along the sequence axis (the second axis), is introduced to globally model the state. This enables transferability across quantum systems of varying sizes, allowing the framework to perform zero-shot transfer learning of circuit representations for quantum states of different sizes.

**Local Fidelity-based Reward Function.** Training based on global fidelity is prone to be trapped by barren plateaus (McClean et al., 2018; Cerezo et al., 2021; Bittel & Kliesch, 2021; Larocca et al., 2024). To address this, we propose a novel reward function based on average local fidelity, inspired by the use of local cost functions to mitigate barren plateaus (Cerezo et al., 2021; Caro et al., 2023). Given two $N$-qubit quantum states $\rho$ and $\sigma$, the average local fidelity is defined as

$$L(\rho, \sigma) = \frac{1}{N}\sum_{i=0}^{N-1} F(\rho_i, \sigma_i), \tag{4}$$

where $F$ is the (global) fidelity between the reduced density matrices $\rho_i$ and $\sigma_i$ of the original states on qubit $i$. This reward is derived exclusively from local measurements. In our scenario, we set $\sigma_i = |0\rangle\langle 0|_i$ and the average local fidelity, denoted as $L(\rho_s^{(t)}, |0\rangle\langle 0|^{\otimes N})$, can be estimated by measuring $\rho_s^{(t)}$ using local operators $\{O_i\}_{i=0}^{N-1}$, where

$$O_i = |0\rangle\langle 0|_i \otimes I_{N\setminus i}, \tag{5}$$

which applies a projector $|0\rangle\langle 0|$ to qubit $i$, and identity to the remaining qubits. The overall operator associated with the average local fidelity is defined as

$$O = \frac{1}{N}\sum_{i=0}^{N-1} O_i. \tag{6}$$

We can compute average local fidelity between the state at step $t$ and the target state as $L(\rho_s^{(t)}, |0\rangle\langle 0|^{\otimes N}) = \text{Tr}(O\rho_s^{(t)})$. The reward for the agent is defined as

$$r^{(t)} = -1 + L(\rho_s^{(t)}, |0\rangle\langle 0|^{\otimes N}). \tag{7}$$

An additional $-1$ term is added into the reward to encourage generating circuits with lower depth. To bound the accuracy of the circuit representation trained with this reward function, we present the following property:

**Property 1** *If the agent learns a policy that constructs an $N$-qubit quantum state with average local fidelity $L(\rho_s^{(T)}, |0\rangle\langle 0|^{\otimes N}) \geq 1 - \epsilon$, then the global fidelity between $\rho_s^{(T)}$ and $|0\rangle\langle 0|^{\otimes N}$ satisfies $F(\rho_s^{(T)}, |0\rangle\langle 0|^{\otimes N}) \geq 1 - N\epsilon$.*

This indicates that high global fidelity can be guaranteed if the agent obtains an effective policy using the defined reward function. The proof is given in Appendix C.

## 3 EXPERIMENTS

In this section, we apply our framework to learn circuit representations for three different families of states – the states prepared by Instantaneous Quantum Polynomial (IQP) circuits, states evolved by Ising Hamiltonians, and quantum many-body ground states. In addition, we use Hamiltonian learning as an example to showcase the interpretability of circuit representations learned by our model. Further discussion on the finite sampling condition and the impact of circuit noise on the performance of our framework can be found in Appendix G.

Our framework is compared with one classical method – Classical Shadow (Huang et al., 2020), one neural network method – Transformer Quantum State (TQS) (Zhang & Di Ventra, 2023), and three quantum methods – Variational Quantum Eigensolver (VQE) (Peruzzo et al., 2014), Quantum Approximate Optimization Algorithm (QAOA) (Farhi et al., 2014) and Quantum Architecture Search (QAS) (Du et al., 2022). The metrics for evaluation are square root global fidelity, second-order Rényi entropy (Rényi, 1961), two-point correlations (Fetter & Walecka, 2003) and spin-Z values (Atkins & de Paula, 2010). The definitions of them are introduced in Appendix D. For the latter three metrics, we compute the Root Mean Squared Error (RMSE) between the true values measured form the target states, and the actual values obtained form the learned representations / output states. For fair comparisons, the training objective for all methods is fidelity—global fidelity for the other methods and local fidelity reward for ours. The properties are predicted without fine-tuning.

### 3.1 LEARNING QUANTUM STATES GENERATED BY INSTANTANEOUS QUANTUM POLYNOMIAL CIRCUITS

IQP circuits are frequently used to benchmark the classical simulatability of quantum circuits (Bremner et al., 2010). While general IQP circuits are classically intractable to simulate, in this experiment, we focus on a specific family of states generated from a discrete gate set to demonstrate the capabilities of our framework. We first apply our framework to construct circuit representations for a family of quantum states prepared by IQP circuits. The output states generated by IQP circuits are

$$|\psi\rangle_k = \bigotimes_{i=0}^{N-1} H_i Z[\alpha]_k \bigotimes_{i=0}^{N-1} H_i |0\rangle^{\otimes N}, \tag{8}$$

where $Z[\alpha]_k$ are single- or two-qubit gates that can be diagonalized in the computational basis, e.g., $Z$, $CZ$ and $R_z(\alpha)$. In our setting, $Z[\alpha]_k$ contains one column of $CZ$ gates acting on every two adjacent qubits, followed by one column of single-qubit gates randomly selected from $\{R_z(\pi/4), R_z(-\pi/4)\}$ for each qubit. We consider a quantum system with a size of $N = 50$. We generate 100 different circuits and record the output states as our training set. The quantum circuits are discarded once the states are generated. We train our framework to reconstruct the circuits that prepare the target states in the training set. The action space for generating circuit representations is $\{H, CZ, R_z(\pi/4), R_z(-\pi/4)\}$. During training, we set the maximum number of iterative steps as $T = 100$. In each step, one gate is applied to one qubit or two nearest neighbor qubits. After training, we generate another 10 different states for evaluation. We use global fidelity and local fidelity

as metrics between the reconstructed states and the target states to test the learned circuit representations. Figure 2(a) and (b) show the scaling of global and local fidelity at step $t$. For 4-qubit states, the local fidelity increases concurrently with global fidelity. However, for 50-qubit states, while the local fidelity monotonically increases, the global fidelity remains stable and sharply rises at the end of the period, highlighting the existence of barren plateaus. Ultimately, the output states of our learned circuits can achieve a fidelity of 0.9999 with the target states on average. Furthermore, we apply our framework trained on 50-qubit systems directly to systems of different scales $N \in \{4, 10, 30\}$ without fine-tuning. Our framework's zero-shot transfer learning performance across these system sizes achieves an average fidelity of 0.9999 in all cases. We also compare the performance of our

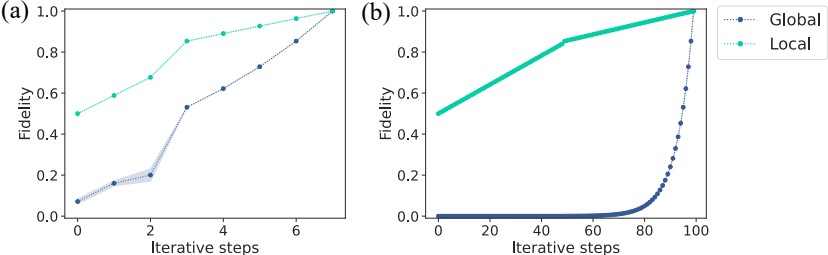

Figure 2: Learning quantum states generated by IQP circuits. (a) States generated by 4-qubit IQP circuits. (b) States generated by 50-qubit IQP circuits.

framework with other state characterization methods for quantum systems of size $N = 4$. The results in Table 2 show that our framework outperforms all others across all metrics.

Table 2: Evaluation results of learning states generated by 4-qubit IQP circuits.

| Method | Metric | Fidelity ↑ | Rényi Entropy ↓ | Two-point Correlations ↓ | Spin-Z ↓ |
|---|---|---|---|---|
| Classical Shadow | 0.9664±0.1025 | 0.4828 | 0.2613 | 0.3699 |
| TQS | 0.6894±0.2946 | 0.5765 | 0.2906 | 0.2309 |
| VQE | 0.9174±0.1042 | 0.1665 | 0.1100 | 0.1539 |
| QAOA | 0.8336±0.1617 | 0.2538 | 0.1026 | 0.1429 |
| QAS | 0.4694±0.1500 | 0.3977 | 0.0895 | 0.2401 |
| **Ours** | **0.9999±0.0001** | **1.06e-06** | **1.80e-07** | **1.98e-07** |

## 3.2 LEARNING QUANTUM STATES EVOLVED BY TRANSVERSE FIELD ISING HAMILTONIANS

Next, we consider learning the circuit representations for a family of states evolved by transverse field Ising Hamiltonians, where the exact parameters of the Hamiltonians and evolution time are agnostic to the framework. Starting with product state $|0\rangle^{\otimes N}$, the state is evolved by an Ising Hamiltonian for time $t$. The target states after the evolution are defined as

$$|\psi\rangle_k = e^{-iH_{\text{Ising}}t}|0\rangle^{\otimes N}, \tag{9}$$

where $H_{\text{Ising}} = J\sum_{i=0}^{N-2} Z_i Z_{i+1} + g\sum_{i=0}^{N-1} X_i$ is the transverse field Ising Hamiltonian, $t$ is the evolution time. In our experiment, we set $N = 50$, $J = -1$, $g \in [-2.0, -1.0]$ and $t \in [0.1, 1.0]$. We sample 10 different $g$s and 10 $t$s uniformly from the range with stride 0.1 to construct the training set of size 100. For training, we set the maximum number of iterative steps $T = 100$, each corresponds to applying one gate to each qubit or every two nearest neighbor qubits. The quantum gates composing the action space are $\{\exp(-i\phi X), \exp(-i\phi Z \otimes Z)\}$. To exhibit the results, we average the performance on different parameters $g$ for each evolution time $t$. Figure 3(a) shows that the learned circuit can successfully recover the target quantum states with high fidelity. Additionally, we evaluate the circuit depth and compare it to the first-order Trotter decomposition (Suzuki, 1985), which is considered one of the most straightforward methods for simulating the dynamics of quantum systems. As shown in Figure 3, our framework can construct circuits shallower than those generated by the Trotter decomposition in general. This indicates that our framework can serve as an optimization technique for traditional quantum simulation technologies. Notably, our framework does not require prior knowledge on the Hamiltonian parameters, offering greater flexibility

compared to the Trotter decomposition method when simulating the dynamics of quantum systems. Figure 3(c) shows the zero-shot transfer performance of applying the framework trained on 50-qubit systems to other quantum $N$-qubit systems with $N \in \{10, 30, 70, 100\}$. The output states remain high fidelity with the target states of unseen sizes, demonstrating the success of our measurement feature aggregation block. In addition, we compare the performance of our framework with other

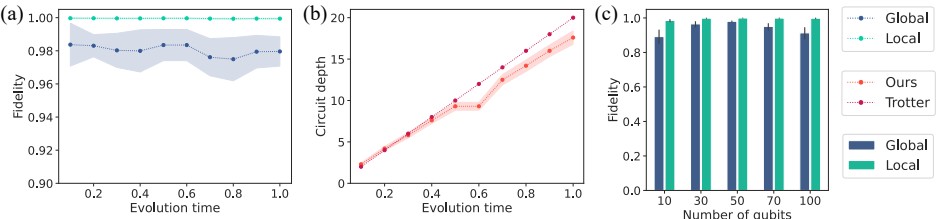

Figure 3: Learning 50-qubit quantum states evolved by Ising Hamiltonians. (a) Scaling of global and local fidelity w.r.t. the evolution time. (b) Comparison of the circuit depths for simulating the dynamics between our framework and the Trotter decomposition method. (c) Zero-shot transfer performance on quantum systems of various sizes. Our framework is trained on the 50-qubit system.

methods for quantum systems of size $N = 4$. Table 3 illustrates the results of predicting different properties. Our framework outperforms other methods on all metrics.

Table 3: Evaluation results of learning 10-qubit states evolved by Ising Hamiltonians, where the evolution time $t \in [0.1, 1]$.

| Method | Metric | Fidelity ↑ | Rényi Entropy ↓ | Two-point Correlations ↓ | Spin-Z ↓ |
|---|---|---|---|---|
| Classical Shadow | 0.9780±0.0332 | 1.4150 | 1.0898 | 2.6851 |
| TQS | 0.8524±0.0957 | 0.1727 | 0.1037 | 0.0944 |
| VQE | 0.2795±0.2359 | 0.5824 | 0.3044 | 0.3619 |
| QAOA | 0.9637±0.1402 | 0.0324 | 0.0382 | 0.0513 |
| QAS | 0.5215±0.2153 | 0.3729 | 0.4349 | 0.4806 |
| Ours | **0.9979±0.0012** | **0.0108** | **0.0227** | **0.0231** |

### 3.3 LEARNING MANY-BODY GROUND STATES

Our third experiment is learning the circuit representations for a family of many-body ground states. We consider two families of ground states separately, the transverse-field Ising Hamiltonian ground states, and the anisotropic Heisenberg XXZ Hamiltonian ground states.

**Learning transverse-field Ising ground states.** In this experiment, we consider the same Ising Hamiltonians as the state evolution experiment in the previous section, but with the goal of learning the ground states rather than time-evolved states. The configurations are $N = 50$, $J = -1$ and $g \in [-2.0, -1.5]$. We uniformly sample 20 different parameters $g$ and compute the corresponding ground states, storing them into the training set. Then we use the QCrep agent to learn the circuits to prepare these ground states. The action space is the same as described in Section 3.2. We set the maximum number of iterative steps $T = 200$ during training. Figure 4(a) shows the scaling of global and local fidelity with the parameters $g$. The local fidelity almost remains stable but the global fidelity slightly drops with the increment of $g$. This indicates that in high-dimensional space, global fidelity is more sensitive to differences between states compared to local fidelity. Thus, global fidelity may not be a good guidance on learning quantum states, in which a relaxed metric encourages exploration and increases the chance of finding the optimal result. Figure 4(c) shows the zero-shot transfer performance of the framework trained on the 50-qubit system when applied to system sizes of $\{10, 30, 70, 100\}$. The comparison results between different methods for learning 10-qubit Hamiltonian ground states are presented in Table 4.

**Learning anisotropic Heisenberg XXZ ground states.** Here, we learn the circuit representations of ground states of a family of 1-D Heisenberg XXZ Hamiltonians. The Hamiltonian is

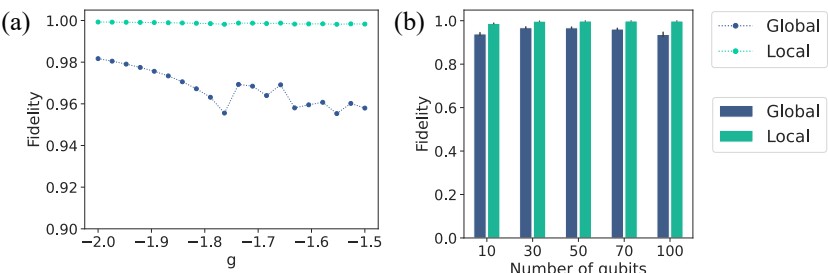

Figure 4: Learning 50-qubit ground states of transverse-field Ising model. (a) Scaling of global and local fidelity w.r.t. Ising parameters. (b) Zero-shot transfer performance on learning Ising ground states of various sizes. Our framework is trained on the 50-qubit system.

Table 4: Evaluation results of learning ground states of 10-qubit Ising systems.

| Method \| Metric | Fidelity ↑ | Rényi Entropy ↓ | Two-point Correlations ↓ | Spin-Z ↓ |
|---|---|---|---|---|
| Classical Shadow | **0.9751±0.0437** | 1.2751 | 1.0431 | 3.9791 |
| TQS | 0.9537±0.0724 | 0.1187 | 0.0958 | 0.0306 |
| VQE | 0.4773±0.0087 | 0.6442 | 0.1475 | 0.0425 |
| QAOA | 0.9614±0.0181 | **0.0368** | 0.1009 | **0.0229** |
| QAS | 0.8032±0.0450 | 0.2260 | 0.1806 | 0.1706 |
| Ours | 0.9691±0.0083 | 0.0989 | **0.0947** | 0.0309 |

$H_{\text{Heisenberg}} = \sum_{i=0}^{N-1} J_x X_i X_{i+1} + J_y Y_i Y_{i+1} + J Z_i Z_{i+1}$. Throughout the experiment, we set $J_x = J_y = -1$, and $J \in [-3.0, -2.0]$. To construct the training set, we uniformly sample 10 different $J$ and generate the ground state of system size $N = 10$. The action space of the agent is $\{\exp(-i\phi X \otimes X), \exp(-i\phi Y \otimes Y), \exp(-i\phi Z \otimes Z)\}$. We set the maximum iterative steps $T = 100$. The scaling of global and local fidelity with parameter $J$ is shown in Figure 5(a). Besides, we evaluate the trained framework on out-of-distribution data. We generate 9 different ground states corresponding to $J \in [-1.9, -1.1]$, and use the trained framework to generate circuit representations to reproduce these states. Results in Figure 5(b) show that our framework can successfully be generalized to prepare unseen states within the same state family. The comparison with other

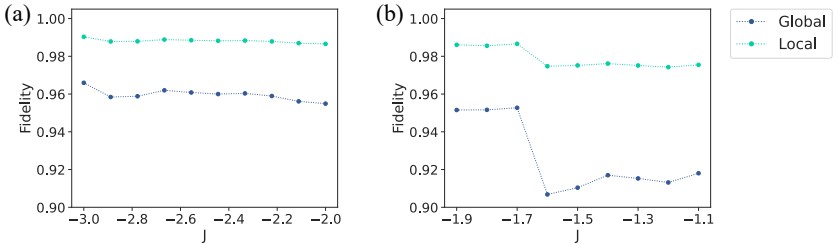

Figure 5: Learning 10-qubit Heisenberg ground states. (a) Scaling of global and local fidelity w.r.t. the parameters. (b) Out-of-distribution generalization.

methods on 10-qubit system is shown in Table 5. Our framework can accurately recover the three properties of the target states and achieves the highest performance among the compared methods.

### 3.4 DOWNSTREAM APPLICATION: HAMILTONIAN LEARNING

After learning the circuit representations for quantum states, it would be interesting to further explore the interpretability of the representations. Here we use Hamiltonian learning as one downstream task to show the effectiveness of the learned circuit representations. Hamiltonian learning is a task to determine the coefficients of an unknown Hamiltonian.

Table 5: Evaluation results of learning ground states of 10-qubit Heisenberg XXZ systems.

| Method \| Metric | Fidelity ↑ | Rényi Entropy ↓ | Two-point Correlations ↓ | Spin-Z ↓ |
|---|---|---|---|---|
| Classical Shadow | 0.9410±0.0408 | 1.3864 | 2.0985 | 1.0000 |
| TQS | 0.6288±0.1204 | 0.7071 | 0.0017 | 0.0159 |
| VQE | 0.4765±0.0105 | 0.0042 | 0.0816 | 0.7840 |
| QAOA | 0.5970±0.0085 | 0.0038 | 0.0693 | **0.0000** |
| QAS | 0.7613±0.0609 | 0.3379 | 0.1234 | 0.4962 |
| Ours | **0.9550±0.0229** | **0.0000** | **0.0000** | **0.0000** |

In our setting, we use the circuit representations of the ground states to learn the corresponding Hamiltonians. The quantum systems we consider are the Ising model and the Heisenberg XXZ model. Specifically, we first use QCrep to learn the circuit representations $(v_t^\dagger, \phi_t)_{t=T}^1$ for ground states corresponding to Hamiltonians with unknown parameters. Next, we concatenate the representations into vectors and pad 0s at the end to ensure the same length. Finally, we employ linear regression to establish the relationship between circuit representations and Hamiltonian parameters using a small training set, and we utilize the learned framework to predict the relationship on the test set. Experimental results in Figure 6 show that, given the circuit representation of a ground state associated with a Hamiltonian with unknown parameters, these unknown parameters can be accurately predicted using only linear regression. Meanwhile, for comparison, we use the circuit

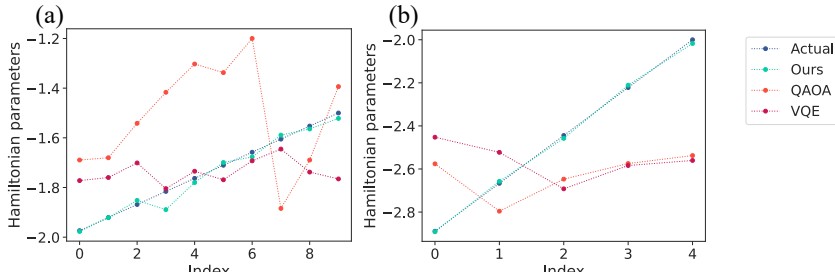

Figure 6: The test set performance of different methods on learning Hamiltonian parameters for 10-qubit (a) Ising and (b) Heisenberg XXZ quantum systems. The x-axis represents the parameter indices, and the y-axis shows the corresponding parameter values.

parameters learned from VQE and QAOA to perform Hamiltonian learning. However, the linear model fails to establish a relationship between the Hamiltonian and circuit parameters. We attribute this outcome to the QCrep learning pipeline, which effectively encodes information about the underlying Hamiltonian into the circuit parameters. This is not achievable with VQE or QAOA, as they rely on gradient-based optimization of the circuit parameters, which perturbs the parameters and hinders the preservation of Hamiltonian information.

## 4 CONCLUSION

We propose a novel type of representations of quantum states – the explicit circuit representations, which feature efficient learning and experimental reconstruction of quantum states. To learn this representation, we design a reinforcement learning framework featuring a Transformer feature aggregation block and a novel local fidelity reward function. The learning procedure relies exclusively on local measurement data, but can recover the target states with high global fidelity. The learned representations can further be transferred to quantum systems of varying sizes and applied to Hamiltonian learning as a downstream task using a linear model.

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

# A RELATED WORK

**Tomography-based quantum state characterization.** Tomography-based methods use direct measurement to characterize quantum states. To accurately characterize the full quantum state, Quantum State Tomography (Tóth et al., 2010; Gross et al., 2010; Cramer et al., 2010; Lanyon et al., 2017; Cotler & Wilczek, 2020) has been proposed, which measures the state in exponential number of basis to obtain the state vector. Other methods focus on constructing a partial knowledge of the state. For instance, Shadow Tomography (Aaronson, 2018) targets at characterizing the measurement values of 2-outcome measurements using only a few copies of the states. Classical shadow (Huang et al., 2020; Akhtar et al., 2023) utilizes randomized measurement to efficiently estimate local properties of the states. Noteworthy, there is a special family of work that uses Tensor network, e.g., Matrix Product State (MPS) (Perez-Garcia et al., 2007) and Projected Entangled Pair States (PEPS) (Scarpa et al., 2020), to approximate the state vector of a quantum state. The original high dimensional state vector is decomposed into multiple low-rank tensors with restricted bound dimension.

**Variational-based quantum state characterization.** Alternative to state tomography, variational quantum algorithms optimize the parameters of a variational ansatz, i.e., a parameterized quantum circuit, to approach the target state. Two representative methods are Variational Quantum Eigensolver (VQE) (Peruzzo et al., 2014) and Quantum Approximate Optimization Algorithm (QAOA) (Farhi et al., 2014). These methods update their output towards the target states, usually the ground states of a Hamiltonian, by measuring the energy and computing quantum gradient descend via, e.g., parameter shift rule (Mitarai et al., 2018). In addition to optimizing parameters, Quantum Architecture Search has been proposed to optimize the circuit ansatz. Du et al. (2022) traverse a candidate gate set and select the gate configurations that achieve the highest scores on the target objective. Wauters et al. (2020); Yao et al. (2021); Ostaszewski et al. (2024) utilize reinforcement learning to optimize the circuit while keeping quantum gradient descend to update parameters. Zhang et al. (2022); Wu et al. (2023a) propose differentiable strategy to simultaneously update the ansatz and parameters.

**Machine learning-based quantum state characterization.** Machine learning can be used to learn the measurement values of states, and predict state properties. The machine learning state characterization methods can mainly be categorized into two classes – Neural Quantum State (Carleo & Troyer, 2017; Sharir et al., 2020; Zhang & Di Ventra, 2023; Chen & Heyl, 2024) and Neural State Representation (Zhu et al., 2022; Tang et al., 2024a; Qian et al., 2024). The Neural Quantum State represents a quantum state as a neural network, where sampling the neural network corresponds to measuring the state. Parameters of the neural network can be updated via Variational Monte Carlo (McMillan, 1965) and Stochastic Reconfiguration (Sorella, 1998) methods. The Neural State Representation compresses the quantum state into a classical description, usually a low dimensional vector, via pretraining. Zhu et al. (2022) adopt a self-supervised manner to predict the measurement values of some measurement operators given other operators. Tang et al. (2024a) use language modeling (Bengio et al., 2003) as the pretraining strategy. In Qian et al. (2024), the vector pretrains the representation by fitting the inner product to fidelity. Afterwards, the pretrained representation can be fine-tuned for downstream tasks, such as predicting the properties of quantum states.

Different from previous machine learning-based methods, we decode the state representation into a novel circuit representation instead of low dimensional vector to support experimental reconstruction ability. Our representation is suitable for downstream applications like Hamiltonian learning. Unlike the reinforcement learning for quantum architecture search, our framework circumvents the need of calculating gradients with respect to the circuit parameters, and possesses the ability to characterize a family of states rather than one specific state.

# B PRELIMINARIES

We review some of the key concepts in quantum computation. For a more comprehensive overview, please refer to Nielsen & Chuang (2010).

Quantum states are quantum counterparts of classical bits. They can be mathematically represented as vectors in Hilbert space, i.e., state vectors, denoted as $|\psi\rangle \in \mathbb{C}^{2^N}$, satisfying $\||\psi\rangle\|_2 = 1$, where $N$ is the system size or the number of qubits. The notation $|\cdot\rangle$ is just used to emphasize that $\psi$ is a (column) vector. Its dual (row vector) is given by $\langle\cdot| \equiv |\cdot\rangle^\dagger$, where "$\dagger$" is the notation for conjugate

transpose. The standard basis for quantum states is the computational basis $\{|i\rangle\}_{i=0}^{2^{N-1}}$, where $|i\rangle$ is the vector whose $i$-th element is 1 and others are 0. For example, $|0\rangle = (1, 0, 0, \cdots, 0)$. Alternatively, we can use the mixed state to describe a probability ensemble of quantum states $\{p_i, |\psi_i\rangle\}$. $p_i$ is the probability of the quantum system being in the state $|\psi_i\rangle$. This can be represented as density matrix $\rho \in \mathbb{C}^{2^N \times 2^N}$, where $\rho = \sum_i p_i |\psi_i\rangle\langle\psi_i|$. Clearly, for pure state $|\psi\rangle$, the corresponding density matrix is $|\psi\rangle\langle\psi|$. Multiple quantum states can be combined to form a compositional system, which is represented by the tensor product (Kronecker product) denoted as "$\otimes$". For two states $|\psi\rangle, |\phi\rangle \in \mathbb{C}^{2^N}$, their composition is given by $|\psi\rangle \otimes |\phi\rangle \in \mathbb{C}^{2^{2N}}$. We use the notation $|\cdot\rangle^{\otimes N}$ to denote an N-qubit product state, e.g., $|0\rangle^{\otimes N} \equiv |0\rangle \otimes \cdots \otimes |0\rangle$.

The similarity between two quantum states can be quantified by (global) fidelity and trace distance. In this paper, we focus exclusively on the global fidelity. Given two density matrices $\rho$ and $\sigma$, the global fidelity is defined as

$$F(\rho, \sigma) = \left( \text{Tr} \left( \sqrt{\rho^{1/2} \sigma \rho^{1/2}} \right) \right)^2. \tag{10}$$

If the two states are pure states $|\psi\rangle$ and $|\phi\rangle$, the fidelity simplifies to $|\langle\psi|\phi\rangle|^2$, which is closely related to the cosine similarity between two vectors.

Quantum states can be measured, causing them to collapse into classical bits. Measurement is described by a set of measurement operators $\{M_j\}$, where each $M_j$ is a Hermitian matrix, i.e., $M_j^\dagger = M_j$. In the case of projective measurements, the operators are projectors that satisfy $\sum_j M_j = I$ and $M_j M_k = \delta_{j,k} M_j$. The measurement outcomes, which correspond to classical bits, are associated with the index $j$. When measuring a state $\rho$, the probability of obtaining outcome $j$ is given by $p(j) = \text{Tr}(M_j \rho)$. The observable $M = \sum_j j M_j$ describes the overall measurement results, and the expectation value of the measurement on the state $\rho$ is $m = \sum_j j p(j) = \text{Tr}(M\rho)$. Additionally, measurement operators can be composed using tensor products to form new measurements for larger quantum systems.

Quantum states can be evolved by quantum gates, analogous to classical logical gates, which are represented by unitary matrices $U$ that satisfy $U^\dagger U = UU^\dagger = I$. A unitary matrix can be generated from a Hamiltonian $H$ – a Hermitian matrix – using a parameter $\phi$, and is expressed as $U(\phi) = \exp(-iH\phi)$. A special group of unitary matrices are Pauli matrices – $X$, $Y$, and $Z$, where

$$X = \begin{bmatrix} 0 & 1 \\ 1 & 0 \end{bmatrix}, \qquad Y = \begin{bmatrix} 0 & -i \\ i & 0 \end{bmatrix}, \qquad Z = \begin{bmatrix} 1 & 0 \\ 0 & -1 \end{bmatrix}. \tag{11}$$

The Pauli matrices form the single-qubit Pauli gates. Besides these gates, other typical quantum gates are single-qubit rotation gates $R_x(\theta) = \exp(-iX\theta/2)$, $R_y(\theta) = \exp(-iY\theta/2)$, $R_z(\theta) = \exp(-iZ\theta/2)$, and two-qubit gates $CX = |0\rangle\langle0| \otimes I + |1\rangle\langle1| \otimes X$, $CZ = |0\rangle\langle0| \otimes I + |1\rangle\langle1| \otimes Z$. More general quantum gates can be decomposed into these single-qubit and two-qubit gates.

## C    PROOF OF PROPERTY 1

Property 1 states that if the agent learns a policy that constructs an $N$-qubit quantum state with average local fidelity $L(\rho_s^{(T)}, |0\rangle\langle0|^{\otimes N}) \geq 1 - \epsilon$, then the global fidelity satisfies $F(\rho_s^{(T)}, |0\rangle\langle0|^{\otimes N}) \geq 1 - N\epsilon$. The proof is given as follows.

**Lemma 1** *Let $k \in \{0, 1, \ldots, N\}$. The operator $O$ has eigenvalues $\lambda_k = 1 - k/N$, where the corresponding algebraic multiplicity is $\binom{N}{k}$.*

Proof. The local operator $O_i$ acting on the $i$-th qubit can be expressed as

$$O_i = \underbrace{I \otimes \cdots \otimes I}_{i} \otimes |0\rangle\langle0|_i \otimes \underbrace{I \otimes \cdots \otimes I}_{N-i-1} \tag{12}$$

$$= \text{diag}(\underbrace{1, 1, \ldots, 1}_{2^i}, \underbrace{0, 0, \ldots, 0}_{2^i}) \otimes \text{diag}(\underbrace{1, 1, \ldots, 1}_{2^{N-i-1}}) \tag{13}$$

$$= \text{diag}(\underbrace{\mathbf{1}_{2^i}, \mathbf{0}_{2^i}, \mathbf{1}_{2^i}, \mathbf{0}_{2^i}, \ldots, \mathbf{1}_{2^i}, \mathbf{0}_{2^i}}_{2^{N-i}}). \tag{14}$$

Now that $O_i$ is a diagonal matrix, the elements 1s and 0s are the eigenvalues. Next, we are interested in the eigenvalues of $O$, which is defined as

$$O = \frac{1}{N} \sum_{i=0}^{N-1} O_i. \tag{15}$$

$O$ is also a diagonal matrix and has eigenvalues $s_j$ with corresponding eigenvectors $|j\rangle$, where $0 \leq j \leq 2^N - 1$. Each term $s_j$ is the sum of $N$ items in the corresponding $j$-th position of local operators $O_i$, denoted by $O_i[j]$, namely

$$s_j = \frac{1}{N} \sum_{i=0}^{N-1} O_i[j]. \tag{16}$$

Since $O_i[j]$ are either 1 or 0, the value of $s_j$ depends on the number of 1s of $O_i[j]$. It is obvious that $0 \leq s_j \leq 1$. Furthermore, we can concatenate $O_i[j]$ into a bitstring and construct the following relation:

$$(O_{N-1}[j], O_{N-2}[j], \dots, O_0[j]) = B(2^N - 1 - j), \tag{17}$$

where the left hand side is the bitstring and $B(2^N - 1 - j)$ is the binary representation of integer $2^N - 1 - j$. We can then use the Hamming distance to characterize the number of 1s in binary$(2^N - 1 - j)$.

Denote $d_{\mathrm{H}}(s, t)$ as the Hamming distance between two equal-length binary numbers $s$ and $t$, which computes the number of positions at which the corresponding bits are different. We fix the length as $N$. Define $\mathcal{S}_k = \{s : d_{\mathrm{H}}(s, 0) = k\}$. The set $\mathcal{S}_k$ contains all $N$-bit binary numbers that have exactly $k$ 1s. It is easy to show that the size of the set $|\mathcal{S}_k| = \binom{N}{k}$. Thus

$$s_j = \frac{1}{N} d_{\mathrm{H}}(B(2^N - 1 - j), 0), \tag{18}$$

and there are $\binom{N}{d_{\mathrm{H}}(B(2^N-1-j),0)}$ repeated $s_j$s. The numbers $d_{\mathrm{H}}(B(2^N - 1 - j), 0)$ take every integer values from 0 to $N$. Sorting $s_j$ in descending order, we can conclude that the eigenvalues are $\lambda_k = 1 - k/N$, with algebraic multiplicity $\binom{N}{k}$.

Denote the eigenvector that corresponds to $\lambda_k$ as $|\lambda_k\rangle$. Let $k = 0$, we have a unique eigenvalue $\lambda_0 = 1$. This is exactly $s_0$ in Equation 18. Therefore, $|\lambda_0\rangle = |0\rangle^{\otimes N}$. Now we construct the relation between average local fidelity $\mathrm{Tr}(O\rho)$ and fidelity $F$ as follows

$$\mathrm{Tr}(O\rho) = \mathrm{Tr}\left(\sum_{k=0}^{N} \lambda_k |\lambda_k\rangle\langle\lambda_k|\rho\right) \tag{19}$$

$$= \langle 0|^{\otimes N} \rho |0\rangle^{\otimes N} + \mathrm{Tr}\left(\sum_{k=1}^{N} \lambda_k |\lambda_k\rangle\langle\lambda_k|\rho\right) \tag{20}$$

$$= F + \sum_{k=1}^{N} \lambda_k \langle\lambda_k|\rho|\lambda_k\rangle \tag{21}$$

$$\leq F + \lambda_1 \sum_{k=1}^{N} \langle\lambda_k|\rho|\lambda_k\rangle \tag{22}$$

$$= F + \lambda_1(1 - \langle\lambda_0|\rho|\lambda_0\rangle) \tag{23}$$

$$= F + \lambda_1(1 - F). \tag{24}$$

Lemma 1 tells us that $\lambda_1 = 1 - 1/N$. Suppose $L(\rho, |0\rangle\langle 0|^{\otimes N}) = \mathrm{Tr}(O\rho) \geq 1 - \epsilon$, then $F \geq 1 - \frac{\epsilon}{1-\lambda_1} = 1 - N\epsilon$.

Q.E.D.

## D    QUANTUM PROPERTIES OF INTEREST IN OUR EXPERIMENTS

In the experiment section, we consider 3 different properties along with global fidelity for performance evaluation, namely the second-order Rényi entropy (Rényi, 1961), two-point correlations (Fetter & Walecka, 2003) and spin-Z values (Atkins & de Paula, 2010). These are important quantities that characterize quantum states from different perspectives. Rényi entropy is a non-linear property, while the two-point correlation and the spin-Z are linear properties.

**Second-order Rényi entropy.** This quantity is used to characterize the subsystem (some of the qubits) entanglement of a quantum state. Denote $\rho_A$ as the reduced density matrix of quantum state $\rho$ on its subsystem $A$, i.e., $\rho_A = \text{Tr}_A(\rho)$. The Rényi entropy quantifies the entanglement strength of $A$, which is computed by

$$\mathcal{S}_\alpha(\rho_A) = \frac{1}{1-\alpha} \log \text{Tr}(\rho_A^\alpha), \tag{25}$$

where $\alpha$ is the order, which is set to 2 in our experiments. We consider the average value of $N-1$ qubit subsystems.

**Two-point correlation.** The correlation function describes the relationships between different parts of the quantum system. This is useful for characterizing quantum phases of matter (Sachdev, 2012) and studying critical behavior (Sachdev, 1999). We consider the two-point correlation defined as follows

$$\mathcal{C}_{0,j} = \text{Tr}(Z_0 Z_j \rho). \tag{26}$$

We take the average of all correlation values for $0 \le j < N$.

**Spin-Z value.** This quantity describes the angular momentum of a many-body quantum state. In our experiments, we consider the spin-Z value, namely the angular momentum in the Z direction, which is defined as

$$s = \text{Tr}\left(\sum_i Z_i \rho\right). \tag{27}$$

To evaluate the performance of different methods in predicting the aforementioned properties, we first compute the true properties of the target states. Next, we apply the benchmarked methods to predict these properties. Finally, we calculate the root mean squared error (RMSE) between the actual and predicted properties as the evaluation metric.

## E    SIMULATION OF QUANTUM SYSTEMS

To simulate large-scale quantum systems, we use the Matrix Product State (MPS) (Perez-Garcia et al., 2007) to represent quantum states, rather than directly using the full state vector. MPS decomposes the state vector into a chain of low-rank tensors through methods such as singular value decomposition, truncating the singular values to compress the state from $\mathcal{O}(2^N)$ to $\mathcal{O}(N\chi^2 d)$ scale, where $d$ is the physical dimension (typically 2 for qubit systems), and $\chi$ is the bond dimension, which represents the number of singular values retained and is related to the degree of entanglement. For product states, $\chi = 1$, while for maximally entangled state, $\chi$ scales exponentially with the system size. Since the quantum states we consider exhibit a low degree of entanglement, e.g., the Ising ground states, the Heisenberg ground states, and states prepared by shallow circuits, we restrict $\chi \le 16$ throughout our experiments.

Afterwards, to simulate the evolution of states, we apply Matrix Product Operators (MPO) (Hubig et al., 2017) to MPS. The evolution of quantum states can be viewed as applying unitaries to the states, which is equivalent to applying MPO to MPS. For single-qubit gates, the MPO is simply the gate itself. For multi-qubit gates, the corresponding MPO can be derived through tensor decomposition similar to MPS. To simulate the time evolution of a state $|\psi\rangle$ governed by a Hamiltonian $H = \sum_l H_l$, where $H_l$ are local Pauli terms, we first apply the first-order Trotter decomposition (Suzuki, 1985) to approximate $e^{-iHt}$. This yields

$$e^{-iHt} \approx \prod_{k=i}^{N} \prod_l e^{-iH_l \tau}, \tag{28}$$

where $\tau$ is the time step and $N = t/\tau$. In the Ising evolution experiment, we set $\tau = 0.1$. Following this, we use the Time-Evolving Block Decimation (TEBD) algorithm (White & Feiguin, 2004) to simulate the evolution. The Hamiltonian terms are divided into even and odd components, and a series of brickwork MPOs are applied to the MPS to perform the time evolution.

For simulating the ground states, we use the DMRG algorithm. First, the Hamiltonian is decomposed into MPO. Then each tensor of MPS is iteratively updated, sweeping from left to right and from right to left. For each tensor, Lanczos method (Lanczos, 1950) is applied to find the eigenvalues and eigenvectors, and the tensor is optimized to the eigenvector with the minimum eigenvalue. This procedure is repeated until the energy converges. In our implementation, the MPS is randomly initialized. We set the maximum dimension of Krylov space to be 10, and the maximum sweep steps to be 200. The iteration stops if the energy difference between to updates is smaller than $10^{-4}$. Note that for Hamiltonians with degenerate eigenspace, the ground states found by DMRG can be different for different initialization of MPS and different parameter specification. Therefore, we turn to imaginary-time evolution (Motta et al., 2020) to simulate the Heisenberg ground states, which is steered by TEBD algorithm with the time being an imaginary number. This guarantees deterministic ground states if the initial MPS, the time step $\tau$ and total steps $N$ are fixed. We set the initial MPS to be $|0\rangle$, $\tau = 0.01$ and $N = 10$.

## F RESOURCE REQUIREMENT FOR TRAINING AND INFERENCE OF QCREP

Table 6 and Table 7 detail the resources required in training and inference for each experiment we conducted respectively. "System size" denotes the number of qubits of the target state family. "#iterations" denotes the total number of iterations required for the RL agent to learn the family of states from beginning until convergence, where each iteration is an episode of maximum length $T = 200$ for Ising ground states and 100 for others. "#observables" is the number observables required for measurements in each iteration.

Table 6: Resource requirement for training.

| Experiment | System size | #iterations | #observables |
|---|---|---|---|
| IQP | 50 | 610 | 441 |
| Evolve Ising | 50 | 1240 | 441 |
| Ground Ising | 50 | 1880 | 441 |
| Ground Heisenberg | 10 | 2040 | 81 |

Table 7: Resource requirement for inference.

| Experiment | System size | Circuit depth | #observables |
|---|---|---|---|
| IQP | 50 | 2 | 441 |
| Evolve Ising | 50 | 10 | 441 |
| Ground Ising | 50 | 22 | 441 |
| Ground Heisenberg | 10 | 28 | 81 |

## G DISCUSSION

### G.1 IMPACT OF FINITE SAMPLING

In the experiments, our framework is trained using expectation values of measurement outcome computed via classical simulation, which requires infinite samples of the quantum states. However, real-world experiments only allow sampling the states for finite times. Moreover, quantum states will collapse after measuring, meaning that the same state has to be prepared multiple times. This results in additional state preparation overhead. Therefore, we design this ablation study to test the performance of our framework under finite sampling settings.

The framework we use is first trained on simulation of infinite sampling data $m = \text{Tr}(M\rho)$ given measurement operator $M$ and state $\rho$. At inference time, we use finite measurement shots $k \in \{128, 256, 512, 1024\}$ to obtain the measurement data $\langle m \rangle_k$ as the input to our framework. The results are shown in Figure 7. Inaccurate measurement has nearly no effect on learning IQP circuits, where the action space contains no continuous parameters. For the other three families of states, using only 512 measurement shots is enough for high fidelity reconstruction, demonstrating the effectiveness of our framework.

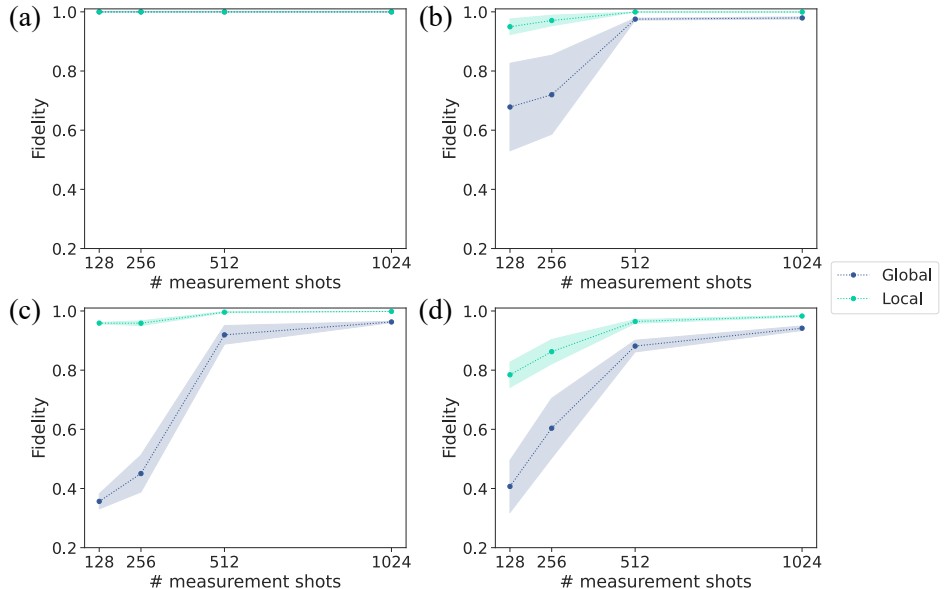

Figure 7: Results under finite sampling conditions. (a) Learning states generated by IQP circuits. (b) Learning states evolved by Ising Hamiltonians. (c) Learning Ising ground states. (d) Learning Heisenberg ground states.

## G.2 IMPACT OF CIRCUIT NOISE

Real-world quantum circuits are affected by noise. Noise causes the actual measurement outcomes biased from the ideal ones. Unlike measurement inaccuracy, this cannot be mitigated via increasing the number of measurement shots. Therefore, it would be interesting to investigate the impact of circuit noise to the construction procedure of circuit representation guided by QCrep.

We evaluate the performance of our framework under the condition that the quantum circuit is affected by a global depolarizing noise. The noisy output state can be formulated as

$$\rho = \mathcal{N}(U|0\rangle\langle 0|U^\dagger), \tag{29}$$

where $\mathcal{N}$ is the noise channel, $U$ is the noise-free circuit. We set the noise parameter associated with the noise strength of $\mathcal{N}$ as $p \in \{0.05, 0.1, 0.15, 0.2\}$. Figure 8 shows the impact of noise strength to global and local fidelity between the learned states and the target states. Even though the fidelity reduces with the increment of noise strength, our framework can maintain a relative good performance within 0.2, demonstrating the robustness of our framework to moderate level of noise. To deal with strong noise, strategies like error correction (Shor, 1995; Fowler et al., 2012) or error mitigation (Giurgica-Tiron et al., 2020; Liao et al., 2023) can be employed to first reduce the noise level. Then the framework can be applied on the low-noise measurement data.

## G.3 UNIVERSAL GATE SET AS ACTION SPACE AND MORE COMPLEX STATE FAMILY

In our experiments, we focus on restricted action spaces. They are constructed by utilizing prior knowledge of the underlying physical system of the target state family. It is an interesting question

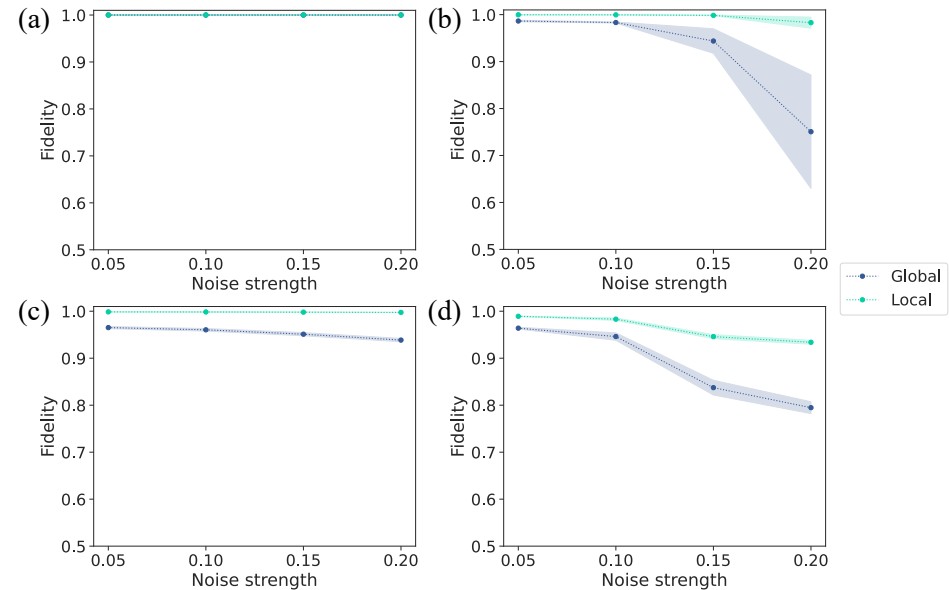

Figure 8: Results under the influence of global depolarizing noise on the quantum circuit. (a) Learning states generated by IQP circuits. (b) Learning states evolved by Ising Hamiltonians. (c) Learning Ising ground states. (d) Learning Heisenberg ground states.

to explore how the agent performs if a universal gate set is considered, and the state family is not restricted to one particular physical system.

Here we consider a mixture state family – the ground states of Ising model together with the ground states of Heisenberg model. The coefficients of the Hamiltonian are chosen the same as in Experiment Section 3.3. We set the number of qubits to be 4. The gate set is chosen as $g = \exp(i\theta G)$, where $G = \{X, Y, Z\} \cup \{X, Y, Z\}^{\otimes 2}$ takes all possible combinations of single- and two-qubit Pauli operators, which form universal 2-local gates. The parameters $\theta \in [-\pi/2, \pi/2]$. Table 8 shows that our model can also perform well using a universal gate set. We highlight that in many practical scenarios, some prior information is available to inform the choice of action space. For example, it is often possible to learn the ground states of a Heisenberg-interaction many-body system without knowing the interaction coefficients but knowing the skeleton of the Hamiltonian.

Table 8: Learning a mixture state family using universal 2-local gates.

| Experiment | System size | Fidelity | Rényi Entropy | Two-point Correlations | Spin-Z |
|---|---|---|---|---|---|
| Mixture family | 4 | 0.9587±0.0130 | 0.0745 | 0.0128 | 0.0434 |

## H    LIMITATIONS

In our measurement settings, we specifically focus on two local Pauli measurements. An interesting future direction would be to explore more universal local measurements and assess whether measuring multiple qubits offers advantages in achieving more accurate circuit construction. Additionally, the neural network we employed in our framework is relatively shallow. Expanding this to a larger framework could enhance expressivity, potentially enabling simultaneous learning states across different quantum phases of matter, and learning ground states of more complex Hamiltonians, e.g., two dimensional Hamiltonians. Besides, it is also interesting to investigate how entanglement affects the performance of our framework, e.g., learn quantum states with volumn-law entangled states that does not allow efficient MPS simulation. Furthermore, for the reinforcement learning algorithm, we have only considered the standard PPO. Incorporating more advanced techniques, such as Monte Carlo Tree Search, could improve training efficiency.

