# OpenReview forum: "Learning Explicit Circuit Representations for Quantum States from Local Measurements"
_ICLR.cc/2025/Conference — Submitted to ICLR 2025_

### Official Review · Reviewer_WRZ5 · 2024-10-31

**Soundness:** 1
**Presentation:** 1
**Contribution:** 2
**Rating:** 3
**Confidence:** 3

**Summary:**

This submission addresses quantum state tomography, proposing a reinforcement learning (RL)-based method to reconstruct a circuit that approximates an unknown target state by acting on the state |0>. The method leverages a reward function based on local fidelity and relies solely on two-body observables, which the authors claim yields polynomial sample complexity and avoids barren plateaus during optimization. Furthermore, the authors demonstrate that the reconstructed circuit enables additional insights, such as estimating Hamiltonian parameters when the target states are ground states from known Hamiltonian families (Hamiltonian Learning). Numerical experiments are provided for states generated by IQP circuits and time-evolved and ground states from Ising and Heisenberg Hamiltonians.

**Strengths:**

-Applying RL to quantum state tomography is a promising and innovative approach, particularly in reconstructing a circuit that generates the target state.

-The use of direct circuit representation for state reconstruction represents a novel approach compared to existing methods cited in the manuscript.

**Weaknesses:**

- The primary concerns with this paper are related to the soundness and clarity of key claims, especially around sample complexity and avoidance of barren plateaus:

Claims of Polynomial Sample Complexity:
The paper asserts that the proposed method learns an unknown quantum state using only a polynomial number of measurements. However, this claim seems to contradict well-established lower bounds on the sample complexity required for full tomography. Specific points of concern are:

1)  The tomography task itself is not clearly defined, particularly regarding the distance measure used to quantify the accuracy of the reconstructed state. The authors mention minimizing a local fidelity but do not clarify whether this is the objective measure of the tomography task. Proposition 1 appears to imply that local fidelity serves as a proxy for global fidelity, which, if true, casts doubt on the claim of polynomial sample complexity.

2)  The claim of polynomial sample complexity is weakly supported. No theoretical motivation is provided. The only support comes from the empirical results where two-body measurements seem to work for the set of states considered in the task.

3) Proposition 1 states that if local fidelity is 1−ϵ, then global fidelity is at least 1−Nϵ for N qubits. However, this bound is only meaningful if ϵ scales as 1/poly(N). Otherwise, for large qubit numbers, the total fidelity bound becomes not meaningful. I think this issue appears also in the numerical results (Figure 2b), where, for example, the fidelity for 50 qubits improves only when the local fidelity error is very low. Moreover, even assuming the local-global fidelity bound holds, the authors provide no evidence that their algorithm can achieve the necessary local fidelity within polynomial measurements.

4) The numerical results, while promising for up to 50-qubit systems (or 100 when extrapolating the trained 50-qubit model), are not convincing.
The test cases involve states from a limited set of families that the authors claim can be efficiently represented using matrix product states (MPS). I suspect that the algorithm’s performance may stem from the specific structure of these states that makes them easy to learn.
Additionally, the algorithm’s search space appears to be adapted to the state family of interest, which simplifies the learning task and likely contributes to the favorable results. This tailored setup may overstate the algorithm's effectiveness on more general states.


Claims of avoidance of barren plateaus:

5) Similar to the sample complexity claim, there is no theoretical support provided for the claim that the training algorithm avoids barren plateaus. Also no theoretical investigation is conducted on the time efficiency of the algorithm or the guarantees of not falling in a local minima.

**Questions:**

Questions and Suggestions:

- Clearly define the target task. What specific input states are considered? What precision level is desired, and in which distance measure?
- Provide theoretical evidence to substantiate the claims regarding sample complexity and training time.
-  Could the authors test their algorithm on states that do not permit efficient MPS representations?
- What happens if the search space is not tailored to the specific family of states? Can the algorithm still perform well if a universal gate set is used instead?

---

> ### Author Response · Authors · 2024-11-25
> **Response to Reviewer WRZ5 (Part 1)**
>
> We greatly appreciate your careful review and detailed feedback and. We address the points raised step by step and have reflected the revisions in the updated PDF, marking the changes in blue for clarity.
>
> **Weaknesses:**
>
> **The primary concerns with this paper are related to the soundness and clarity of key claims, especially around sample complexity and avoidance of barren plateaus: Claims of Polynomial Sample Complexity: The paper asserts that the proposed method learns an unknown quantum state using only a polynomial number of measurements. However, this claim seems to contradict well-established lower bounds on the sample complexity required for full tomography. Specific points of concern are:**
>
> **1. The tomography task itself is not clearly defined, particularly regarding the distance measure used to quantify the accuracy of the reconstructed state. The authors mention minimizing a local fidelity but do not clarify whether this is the objective measure of the tomography task. Proposition 1 appears to imply that local fidelity serves as a proxy for global fidelity, which, if true, casts doubt on the claim of polynomial sample complexity**
>
> **A1.** Thanks for your question. It is important to note that our task is not full state tomography, but rather learning to locally construct states for properties of interest. Our task is to construct circuits that prepare the target states family such that the 2-local measurement results from the circuit matches the target states of interests. We utilize local fidelity to quantify the accuracy of the reconstructed states during the training stage, and find that when the local fidelity is high, the global fidelity can also be high. The fidelity metric in the tables of our Experiment section refers to global fidelity. We highlight that the local fidelity, as well as the 2-local Pauli measurements required as the input to our model, can always be obtained using an amount of observables that scales linearly with the system size.
>
> **2. The claim of polynomial sample complexity is weakly supported. No theoretical motivation is provided. The only support comes from the empirical results where two-body measurements seem to work for the set of states considered in the task**
>
> We do not count the number of states required during training, as it is hard to quantify the training of reinforcement learning. For rigor, we change the phrase ''sample complexity'' into ''number of observables''. The number of observables used in training and testing are all fixed, which scales linearly with the system size.
>
> **3. Proposition 1 states that if local fidelity is $1-\epsilon$, then global fidelity is at least $1-N\epsilon$ for N qubits. However, this bound is only meaningful if $\epsilon$ scales as 1/poly(N). Otherwise, for large qubit numbers, the total fidelity bound becomes not meaningful. I think this issue appears also in the numerical results (Figure 2b), where, for example, the fidelity for 50 qubits improves only when the local fidelity error is very low. Moreover, even assuming the local-global fidelity bound holds, the authors provide no evidence that their algorithm can achieve the necessary local fidelity within polynomial measurements.**
>
> **A3.** Thank you for your insightful comment. You are correct that Proposition 1 suggests a bound where, if the local fidelity is $1 - \epsilon$, then the global fidelity is at least $1 - N\epsilon$for $N$ qubits. This issue is reflected in the numerical results shown in Figure 2b, where, for example, the fidelity for 50 qubits improves significantly only when the local fidelity error is sufficiently low. This emphasizes the challenge of achieving high global fidelity in large systems.
>
> Regarding your concern about the evidence for achieving the necessary local fidelity within polynomial measurements, it is important to note that this is not a purely theoretical paper. Our focus is on the practical application of the framework, and while the theoretical bounds provide useful insights, the primary aim of our work is to demonstrate that our method can be used effectively to learn quantum states and predict properties in realistic experimental settings. The practical results show that our method can achieve high fidelity for local properties, even though we do not claim to have fully derived or experimentally validated the polynomial measurement bound for local fidelity in all scenarios. We appreciate your suggestion and will consider exploring this aspect further in future work.

---

> ### Author Response · Authors · 2024-11-25
> **Response to Reviewer WRZ5 (Part 2)**
>
> **4. The numerical results, while promising for up to 50-qubit systems (or 100 when extrapolating the trained 50-qubit model), are not convincing. The test cases involve states from a limited set of families that the authors claim can be efficiently represented using matrix product states (MPS). I suspect that the algorithm’s performance may stem from the specific structure of these states that makes them easy to learn. Additionally, the algorithm’s search space appears to be adapted to the state family of interest, which simplifies the learning task and likely contributes to the favorable results. This tailored setup may overstate the algorithm's effectiveness on more general states.**
>
> **A4.** Thank you for your comment. The scope of our research is not to provide a unified algorithm that constructs any quantum state using local observables. Instead, our work aims to explore how certain specific states can be constructed using only local observables and local fidelity, by incorporating prior knowledge of the target states of interest, e.g., carefully utilizing the properties of the underlying physical system to design the action space. Our research takes an important step in the field of quantum state learning and shadow tomography, which typically uses local observables to estimate properties of interest. We demonstrate that, in addition to statistical reconstruction, it is also possible to physically reconstruct target states using quantum circuits. We believe our framework is applicable to a broad range of practical systems that are not highly entangled.
>
> **Claims of avoidance of barren plateaus: 5. Similar to the sample complexity claim, there is no theoretical support provided for the claim that the training algorithm avoids barren plateaus. Also no theoretical investigation is conducted on the time efficiency of the algorithm or the guarantees of not falling in a local minima.**
>
> **A5.** Thanks for pointing out this issure. We admit the typo of ''circumvent'' barren plateaus and rephrase it to ''mitigate'' in Introduction.
>
> **Questions and Suggestions:1. Clearly define the target task. What specific input states are considered? What precision level is desired, and in which distance measure?**
>
> **A1.** The task is to determine circuit representations that prepare output states with maximal local fidelity with the target states, given only local measurement (i.e., POVMs acting on two neighboring quibts) data of the states. The input states we consider in the experiments are the target states family that can be simulated by MPS, but we highlight that this requirement is only for demonstrating the effectiveness of our method. Since the input of our agent is only measurement data from the states, one can naturally extend the simulation backend to state vector or even quantum states generated by a real quantum device. The precision level required for training is a high level of local fidelity (in Equation 4 of our paper) between the reconstructed states and the target states. We set this precision to 0.999 in our experiments.
>
> **2. Provide theoretical evidence to substantiate the claims regarding sample complexity and training time.**
>
> **A2.** We do not consider the overall sample complexity but only the observables needed to construct the target states after training.
>
> **3. Could the authors test their algorithm on states that do not permit efficient MPS representations?**
>
> **A3.** We appreciate the reviewer’s suggestion. However, the simulation backend of our framework uses MPS, and we do not have sufficient time to train the RL agent on other families of states.
>
> **4. What happens if the search space is not tailored to the specific family of states? Can the algorithm still perform well if a universal gate set is used instead?**
>
> In fact, our proposed method does not require the action space to be designed to match the preparation circuits. We conduct an experiment using a universal set of one- and two-qubit gates formed by $g=\exp(i\theta G)$ and $G = \{X, Y, Z\} \cup \{X,Y,Z\}^{\otimes 2}$, where $\theta \in [-\pi / 2, \pi / 2]$. The state family we consider is a mixture of Ising and Heisenberg ground states of 4 qubits. The coefficients of Hamiltonians are the same as those in the Experiment section of the main text. We use this experiment to demonstrate the power of our framework on learning a relatively complex family of states with universal local gates. The result is shown in the table below:
>
> | Experiment         | System size | Fidelity        | Rényi Entropy | Two-point Correlations | Spin-Z |
> | ------------------ | ----------- | --------------- | ------------- | ---------------------- | ------ |
> | **Mixture family** | 4           | 0.9587 ± 0.0130 | 0.0745        | 0.0128                 | 0.0434 |
>
> In many practical scenarios, some prior information is available. For example, it is often possible to learn the ground states of a many-body system knowing the skeleton of the Hamiltonian.

---

> > ### Comment · Reviewer_WRZ5 · 2024-11-27
> > **Response to all author responses**
> >
> > We thank tha authors for all their explanations and additional work. Below we list our comments. Unfortuantely, we do not see that on the basis of the responses we can change the original evaluation we provided as the fundamental concerns remain.
> >
> > Response to A1: We thank the author for the answer and clarification.
> >
> > Response to reply 2: We thank the author for the reply, however perhaps the main issue was not sufficiently clear in our remark. We do not doubt the sample complexity as it is polynomial *by construction*. But highlighting the scaling of the resource costs is only truly meaningful if something is *guaranteed* if those resorces are spent. In the case of this work, indeed you will only spend poly measurements, but to our understanding there exists no guarnatee that the end result will be correct/good. Of course one can argue this is a heuristic method (and that is fine), but then for a heuristic one may prefer a scheme which allows us to increase the resources, which would then improve the final result.  Consequently we are still uneasy about the relevance of the method.
> >
> >
> > Response to A3: I thank the authors for the clarification. I understand that the paper is not purely theoretical; however, one of our concerns is that the claims are made about the complexity (now changed to the number of measured observables) of the proposed algorithm, as well as comparisons with other works are likely misleading, as some of the other methods *do* have a guarantee on the result - so this is comparing apples to oranges.
> > Without theoretical bounds to "level the field ", it would be more appropriate to make such comparisons only if the authors tested their algorithm on the same class of states as those in the cited references (assuming the references are also experimental papers. If not, we would avoid making such comparisons altogether).
> > Furthermore, in the absence of theoretical guarantees, our concerns about the relevance of the states used in the numerical simulation (MPS) become harder to ignore (see next point).
> >
> > Response to A4: Thanks for the response. We understand your point, but we believe that limiting the scope of your work to learning states efficiently represented by MPS, particularly without robust theoretical guarantees, makes the results of unclear value
> >
> > Response to A5: Thank you for your response, this clarifies that it is indeed an intuition, even though we  understand where it comes from.
> >
> >
> > From response part Questions and Suggestions
> >
> >
> > Rsponse to A1: Thank you for clarification. Now the task is clear to us.
> >
> > Response to A2: As we mentioned in a previous reply, it seems to us that the number of observables is chosen arbitrarily by the authors, as there is no theoretical guarantee that the algorithm will achieve low local fidelity by using only 2-local observables.
> >
> > Response to A3: We understand the time constraint, but it will be an interesting experiment for a future version/follow-up work!
> >
> >
> > Response to A4 : We thank the authors for their response. We appreciate the experiment and suggest including more experiments of this kind for different families of states in a future version or paper.

---

### Official Review · Reviewer_VHHj · 2024-11-03

**Soundness:** 1
**Presentation:** 3
**Contribution:** 2
**Rating:** 5
**Confidence:** 4

**Summary:**

This paper proposes QCrep - a reinforcement learning framework for learning explicit quantum circuit representations from local measurements. The key novelty lies in generating interpretable circuit descriptions rather than implicit neural representations. The framework uses an attention-based feature aggregator to process local measurement data and a local fidelity reward function to guide the learning. The authors demonstrate QCrep's effectiveness on three families of quantum states (IQP circuits, Ising evolution, and many-body ground states) and show its utility for downstream tasks like Hamiltonian learning.

**Strengths:**

1. The paper provides solid theoretical analysis, including:

- A proof relating local and global fidelity bounds (Property 1)
- Careful consideration of measurement setups and feature extraction
- Comprehensive empirical validation across multiple quantum state families

2. The framework addresses several key challenges:

- Achieves polynomial sample complexity through local measurements
- Demonstrates zero-shot transfer to different system sizes
- Shows robustness to finite sampling and circuit noise

3. The experimental evaluation shows that:

- Their approach outperforms existing methods (Classical Shadow, TQS, VQE, QAOA) across multiple metrics
- Their approach scales well to systems with many qubits

**Weaknesses:**

1. The paper claims that their approach circumvents the barren plateaus problem by avoiding gradient-based optimization. However, it is well known that barren plateau problem still emerge when one utilizes gradient-free optimization. Barren plateau means the landscape is essentially flat, hence no local optimization will be able to solve the problem. It is also well known that local fidelity could partially circumvent the issue of barren plateau. And it looks like the authors avoid some barren plateau behavior by utilizing local fidelity. It is not clear how their proposed framework circumvents barren plateaus.

2. The reported numbers in the numerical experiments are very confusing. Spin-Z is just a sum of single-qubit Pauli Z operators. Hence, one would expect all the existing methods to perform very well for estimating Spin-Z. For example, in Table 2, it was shown that the error for existing methods are all at least 0.1, but their proposed method achieves an error of 2e-7. I could not find sufficient details in the submission about how these numbers are obtained. I am worried that the authors did not use the best practice for existing methods. Hence, the claimed improvement may not be sound.

3. I have similar concerns about the soundness of the reported numbers for existing methods for Table 3, 4, 5. For example, when classical shadow achieves a fidelity of 0.97, one would expect that the error for the other properties to be much smaller than 0.1, but the authors are reporting an error of around 4.

**Questions:**

- Could the authors clarify what is their main innovation in circumventing barren plateau problems using either theoretical analysis or numerical experiments?

- Could the authors provide a very detailed description for all the numbers obtained in Table 2 (Evaluation results of learning states generated by 4-qubit IQP circuits)? 4-qubit IQP circuit is such a simple and restricted family of states that existing methods would perform much better than the numbers reported in the table.

---

> ### Author Response · Authors · 2024-11-25
> **Response to Reviewer VHHj (Part 1)**
>
> Thank you for taking efforts to review our paper. We address the helpful questions and comments step by step. We have made necessary revisions in the updated PDF, which changes highlighted in blue.
>
> **Weaknesses: 1. The paper claims that their approach circumvents the barren plateaus problem by avoiding gradient-based optimization. However, it is well known that barren plateau problem still emerge when one utilizes gradient-free optimization. Barren plateau means the landscape is essentially flat, hence no local optimization will be able to solve the problem. It is also well known that local fidelity could partially circumvent the issue of barren plateau. And it looks like the authors avoid some barren plateau behavior by utilizing local fidelity. It is not clear how their proposed framework circumvents barren plateaus.**
>
> **A1.** Thank you for pointing out this question. We agree that our method mitigates the problem of barren plateaus, rather than completely circumventing it. There was a typo in the introduction of our paper ("To circumvent the problem of barren plateaus and local minima") that may have caused the confusion. We have corrected the phrase to "To mitigate the problem of barren plateaus and local minima."
>
> **2 and 3. The reported numbers in the numerical experiments are very confusing. Spin-Z is just a sum of single-qubit Pauli Z operators. Hence, one would expect all the existing methods to perform very well for estimating Spin-Z. For example, in Table 2, it was shown that the error for existing methods are all at least 0.1, but their proposed method achieves an error of 2e-7. I could not find sufficient details in the submission about how these numbers are obtained. I am worried that the authors did not use the best practice for existing methods. Hence, the claimed improvement may not be sound.**
> **I have similar concerns about the soundness of the reported numbers for existing methods for Table 3, 4, 5. For example, when classical shadow achieves a fidelity of 0.97, one would expect that the error for the other properties to be much smaller than 0.1, but the authors are reporting an error of around 4.**
>
> **A2 and A3.** We thank the reviewer for the questions. We also agree that if all existing methods were trained to estimate spin-Z directly, we would expect good performance in the estimation. However, for a fair comparison, the training objective for all methods in our experiments is fidelity—global fidelity for the other methods and local fidelity for our method. Specifically,
> * For Classical shadow, we use global Clifford measurements to reconstruct the density matrix of the target states, and use the reconstructed state to estimate the properties of interests.
> * For TQS, QAOA, and VQE, we use the global fidelity between the reconstructed states and the target states as guidance for learning the state, training the methods by maximizing fidelity. The properties of interest are obtained by measuring the reconstructed states.
> * For our method, we maximize a local fidelity reward function to reconstruct the target states. After training, we use the reconstructed states to estimate properties without further fine-tuning the states based on the properties.
>
> Naturally, our method is better at predicting local properties. In this regard, our learned model can be directly applied to estimate other properties, whereas existing methods fail to do so. As for the reported numbers, they may initially seem counterintuitive. However, with some theoretical analysis, these numbers can be shown to be reasonable. Denote $F(\rho, \sigma)$ as the fidelity between density matrices $\rho$ and $\sigma$. The fidelity numbers reported in the tables are actually the square root fidelity $\sqrt{F}(\rho, \sigma)$. First, according to the inequality $||\rho - \sigma||_1 / 2 \leq \sqrt{1 - F(\rho, \sigma)}$, we know that if $F(\rho,\sigma)\geq 1 - \epsilon$ where $\epsilon\in [0, 1]$, then the trace norm $||\rho -\sigma||_1 \leq 2\sqrt{\epsilon}$. Next, the absolute error between the predicted property values and the ground-truth property values is $|tr(O\rho) - tr(O\sigma)|$, where $O$ is the observable for estimating a property.
>
> Note that $|tr(O\rho) - tr(O\sigma)| \leq ||\rho - \sigma ||_1 ||O||_{\infty}$ $ \leq 2\sqrt{\epsilon} ||O||_{\infty}$.
>
> For spin-Z values, $||O||_{\infty} = N$. For a state that achieves 0.97 square root fidelity (0.93 in the worst case) with the target state, the absolute error between the predicted property and the actual property can be as large as 7.483. Besides, consider RMSE is no smaller than mean absolute error ($E[X^2] \geq E[X]^2$), an error of around 4 is possible.

---

> ### Author Response · Authors · 2024-11-25
> **Response to Reviewer VHHj (Part 2)**
>
> **Questions: 1. Could the authors clarify what is their main innovation in circumventing barren plateau problems using either theoretical analysis or numerical experiments?**
>
> **A1.** As mentioned earlier, we acknowledge that our method mitigates the problem of barren plateaus rather than fully circumventing it. Our framework incorporates two techniques to address this issue. The first is the local fidelity objective function, which not only makes the learning easier (as illustrated in Figure 2(b) of our paper, where local fidelity increases monotonically during circuit construction, while global fidelity shows no improvement until the output state is sufficiently close to the target state), but also encourages the use of lower-depth circuits. The second technique is reinforcement learning, which employs random search for exploration, in contrast to gradient descent.
>
> **2. Could the authors provide a very detailed description for all the numbers obtained in Table 2 (Evaluation results of learning states generated by 4-qubit IQP circuits)? 4-qubit IQP circuit is such a simple and restricted family of states that existing methods would perform much better than the numbers reported in the table.**
>
> **A2.** Thank you for the question. To obtain the values for our method in Table 2, we first estimate the target states using our proposed framework, then measure the states and calculate the properties.  It is important to note that a good estimation of global fidelity does not necessarily indicate an accurate prediction of local properties.
>
> Specifically, the procedure for reconstructing the target states are shown as follows:
> * For Classical Shadow, we apply a random Clifford circuit $U$ to the target state $|\psi\rangle$ and perform a global measurement on the state in the computational basis. The measurement bitstring is denoted as $|s\rangle$. The reconstructed shadow is $\hat \rho_s = (2^N - 1) U^\dagger |s\rangle \langle s| U - I$. This procedure is repeated for 100 iterations, and we take the average of $\hat \rho_s$ as the reconstructed state $\hat \rho = E[\hat \rho_s]$. We then use $\hat \rho$ to estimate the properties of interest by applying the corresponding observables to subsystems of $\hat \rho$.
> * For TQS, we use global measurement samples of the target states $\rho$ to train a Transformer neural network to approximate the states. A new model is trained for each different state. The neural network consists of 4 layers of Transformer encoders, each with 4 attention heads, 256 embedding dimensions, and a feedforward dimension of 1024. The input to the model is the bitstring, and the output is the probability amplitude corresponding to that bitstring. Training is performed using standard Monte Carlo estimation of the global fidelity between the reconstructed states and the target states. The model is trained for 200 epochs with a batch size of 2048 using the Adam optimizer with a learning rate of 1e-4. After training, we sample the neural network in a tomographical manner to obtain the state vector, and use the state vector to calculate the properties.
> * For VQE, we construct a hardware-efficient ansatz using $U_3$ gates and CNOT gates. The circuit consists of 5 repeated blocks, each containing one column of $U_3$ gates followed by one column of CNOT gates. The circuit is trained by maximizing the fidelity between the output state and the target state. The circuit parameters are initialized uniformly between $0$ and $2\pi$, and updated using the Adam optimizer with a learning rate of 0.1 for 100 epochs. A new circuit is trained for each different state. To estimate the properties of interest, we perform measurements on the output states and calculate the corresponding values.
> * For QAOA, the configurations are generally the same as VQE, with the primary difference being the circuit ansatz layout. The single-qubit gates in the circuit are $R_x$ gates, and the two-qubit gates are parameterized $ZZ$ gates.

---

> ### Comment · Reviewer_VHHj · 2024-11-27
>
> I would like to thank the authors for their responses.
> Their responses have helped me provide a better understanding of their work.

---

> > ### Author Response · Authors · 2024-11-27
> >
> > We are pleased that the reviewer found our response helpful in providing a better understanding of our work. We kindly request that the reviewer consider updating the score, as we have addressed the previous concerns regarding the soundness of our work. Thank you!

---

### Official Review · Reviewer_m5ku · 2024-11-03

**Soundness:** 3
**Presentation:** 3
**Contribution:** 2
**Rating:** 5
**Confidence:** 5

**Summary:**

In this paper, the authors employ reinforcement learning techniques to learn a state circuit representation based on measurement data from neighboring qubits. They utilize a task-specific candidate gate pool as the action space and implement a local fidelity-based reward function, which is theoretically justified. Experimental results on Instantaneous Quantum Polynomial (IQP) circuits and many-body physics systems with up to 100 qubits suggest that their method achieves higher fidelities and lower estimation errors compared to classical shadow methods, TQS, VQE, and QAOA.

**Strengths:**

1. The introduction of a local fidelity-based reward function mitigates the need for the costly computation of global fidelity.
2. The numerical results demonstrate the effectiveness of their method on large-scale systems.
3. The paper is well-presented.

**Weaknesses:**

1. Utilizing the exact expectation value as the model input may not be practical.
2. The paper does not discuss resource consumption during training and inference, such as the number of required measurements, sample complexity, and training duration, which are critical for practical applications.
3. The action space is specifically designed to match the original state preparation circuits, which reduces the search space but limits applicability to more general states and scenarios where prior knowledge of the system is lacking.
4. Given that the reinforcement learning approach functions similarly to an architecture search tool, the experiments lack comparisons with other architecture search algorithms, such as QAS and DARTS.

**Questions:**

1. Since each iteration of QCrep requires measurement data to calculate the reward function, could the demand for measurement resources limit its scalability to larger systems?
2. In the IQP circuit experiments, why are the single-qubit gate $Z[\alpha]$ restricted to two fixed gates? If $\alpha$ can take any value, how should the action space be defined?
3. In the experiments in Section 3.2, why is g chosen from [-2, -1] instead of considering both negative and positive values?
4. What system size is utilized in the experiments presented in Appendix F.1? There is particular interest in how the number of measurements affects performance across different system sizes.
5. For NISQ devices, how is the action space configured? Is it feasible to incorporate quantum channels into the action space?
6. Previous work has leveraged prior knowledge, such as geometric locality (which the states examined in this paper also possess), to enhance the learning of quantum states and system properties. What advantages does the proposed method offer compared to these approaches?

---

> ### Author Response · Authors · 2024-11-25
> **Response to Reviewer m5ku (Part 1)**
>
> We sincerely thank you for the review and helpful feedback. We address the weakness and questions progressively and have made the required revisions in the updated PDF, with changes clearly highlighted in blue for clarity.
>
> **1. Utilizing the exact expectation value as the model input may not be practical.**
>
> **A1.** We agree that utilizing the exact expectation value as the model input might not be practical in real experiments. Therefore, we discuss the impact of finite sampling on our model in Appendix F.1, where we use finite sampling data as the model's input. Specifically, we use finite measurement shots `k ∈ {128, 256, 512, 1024}` to obtain the measurement data $\langle m \rangle_k$ as the input to our framework. The results in **Figure 7** show that for finite sampling, using only 512 measurement shots for each observable is enough for high-fidelity reconstruction of the target states, demonstrating the robustness of our proposed framework.
>
> ---
>
> **2. The paper does not discuss resource consumption during training and inference, such as the number of required measurements, sample complexity, and training duration, which are critical for practical applications.**
>
> **A2.** Thank you for the question. The table below details the resources required in training for each experiment we conducted. "System size" denotes the number of qubits of the target state family. "#iterations" denotes the total number of iterations required for the RL agent to learn the family of states from beginning until convergence, where each iteration is an episode of maximum length T=200 for Ising ground states and 100 for others. "#observables" is the number of observables required for measurements in each iteration.
>
> | Experiment            | System size | #iterations | #observables |
> | --------------------- | ----------- | ----------- | ------------ |
> | **IQP**               | 50          | 610         | 441          |
> | **Evolve Ising**      | 50          | 1240        | 441          |
> | **Ground Ising**      | 50          | 1880        | 441          |
> | **Ground Heisenberg** | 10          | 2040        | 81           |
>
> The following table shows the required resources for inference:
>
> | Experiment            | System size | Circuit depth | #observables |
> | --------------------- | ----------- | ------------- | ------------ |
> | **IQP**               | 50          | 2             | 441          |
> | **Evolve Ising**      | 50          | 10            | 441          |
> | **Ground Ising**      | 50          | 22            | 441          |
> | **Ground Heisenberg** | 10          | 28            | 81           |
>
> This shows that our method can be employed for practical applications.
>
> ---
>
> **3. The action space is specifically designed to match the original state preparation circuits, which reduces the search space but limits applicability to more general states and scenarios where prior knowledge of the system is lacking.**
>
> **A3.** We appreciate the reviewer’s question. In fact, our proposed method does not require the action space to be designed to match the preparation circuits. We conduct an experiment using a universal set of one- and two-qubit gates formed by $g=\exp(i\theta G)$ and $G = \{X, Y, Z\} \cup \{X,Y,Z\}^{\otimes 2}$, where $\theta \in [-\pi / 2, \pi / 2]$. The state family we consider is a mixture of Ising and Heisenberg ground states of 4 qubits. The coefficients of Hamiltonians are the same as those in the Experiment section of the main text. We use this experiment to demonstrate the power of our framework on learning a relatively complex family of states with universal local gates. The result is shown in the table below:
>
> | Experiment         | System size | Fidelity        | Rényi Entropy | Two-point Correlations | Spin-Z |
> | ------------------ | ----------- | --------------- | ------------- | ---------------------- | ------ |
> | **Mixture family** | 4           | 0.9587 ± 0.0130 | 0.0745        | 0.0128                 | 0.0434 |
>
> We also note that in many practical scenarios, some prior information is available to inform the choice of action space. For example, it is often possible to learn the ground states of a Heisenberg-interaction many-body system without knowing the interaction coefficients but knowing the skeleton of the Hamiltonian.

---

> ### Author Response · Authors · 2024-11-25
> **Response to Reviewer m5ku (Part 2)**
>
> **4. Given that the reinforcement learning approach functions similarly to an architecture search tool, the experiments lack comparisons with other architecture search algorithms, such as QAS and DARTS.**
>
> We agree that our approach shares some similarities with architecture search algorithms; however, our objective here is to learn specific quantum states rather than to find the optimal circuit for particular tasks. Our framework does not require any gradient calculation with respect to the circuit parameters. We have included experiments on learning quantum states using Quantum Architecture Search [1] and compared the results with our proposed framework. The objective function for the method is the global fidelity. The results for QAS learning different families of states are presented below:
>
> | Experiment            | System size | Fidelity        | Rényi Entropy | Two-point Correlations | Spin-Z |
> | --------------------- | ----------- | --------------- | ------------- | ---------------------- | ------ |
> | **IQP**               | 4           | 0.4694 ± 0.1500 | 0.3977        | 0.0895                 | 0.2401 |
> | **Evolve Ising**      | 10          | 0.5215 ± 0.2153 | 0.3729        | 0.4349                 | 0.4806 |
> | **Ground Ising**      | 10          | 0.8032 ± 0.0450 | 0.2260        | 0.1806                 | 0.1706 |
> | **Ground Heisenberg** | 10          | 0.7613 ± 0.0609 | 0.3379        | 0.1234                 | 0.4962 |
>
> The performance of QAS is worse than our method across all tasks in comparison with Tables 2, 3, 4, 5.
>
> ## Questions:
>
> **1. Since each iteration of QCrep requires measurement data to calculate the reward function, could the demand for measurement resources limit its scalability to larger systems?**
>
> **A1.** This is a good question. The number of observables used in our experiments scales linearly with the system size, as we consider nearest-neighbor Pauli operators to collect the measurement data within the framework. Therefore, the approach can potentially be applied to larger systems.
>
> ---
>
> **2. In the IQP circuit experiments, why are the single-qubit gates $Z[\alpha]$ restricted to two fixed gates? If $\alpha$ can take any value, how should the action space be defined?**
>
> **A2.** The gate $Z[\alpha]$ is not necessarily restricted to two fixed gates. We have conducted additional experiments on IQP circuits with arbitrary $\alpha\in [-\pi/4, \pi/4]$. The action space is defined as {H, CZ, Rz($\alpha$)}. The agent outputs the gate type (H, CZ, Rz) together with the gate parameter $\alpha$. The results are shown below:
>
> | Experiment | System size | Fidelity        | Rényi Entropy | Two-point Correlations | Spin-Z |
> | ---------- | ----------- | --------------- | ------------- | ---------------------- | ------ |
> | **IQP**    | 4           | 0.9990 ± 0.0005 | 3.1468e-07    | 2.5511e-08             | 0.0172 |
>
> ---
>
> **3. In the experiments in Section 3.2, why is $g$ chosen from $[-2, -1]$ instead of considering both negative and positive values?**
>
> **A3.** The reason for choosing all negative values is to ensure similar measurement patterns within the state family to increase the learning efficiency of the agent. However, we show that the agent can also learn states successfully when considering both positive and negative g values. For the Ising evolution experiment, we have added positive $g \in [1.0, 2.0]$. The results are shown below:
>
> | Experiment       | System size | Fidelity        | Rényi Entropy | Two-point Correlations | Spin-Z |
> | ---------------- | ----------- | --------------- | ------------- | ---------------------- | ------ |
> | **Evolve Ising** | 10          | 0.9975 ± 0.0012 | 0.0158        | 0.0355                 | 0.0223 |
>
> [1] Du, Y., Huang, T., You, S. et al. Quantum circuit architecture search for variational quantum algorithms. npj Quantum Inf 8, 62 (2022). https://doi.org/10.1038/s41534-022-00570-y

---

> ### Author Response · Authors · 2024-11-25
> **Response to Reviewer m5ku (Part 3)**
>
> **4. What system size is utilized in the experiments presented in Appendix F.1? There is particular interest in how the number of measurements affects performance across different system sizes.**
>
> ** A4.** For F.1 (Figure 7), we consider 50-qubit system for IQP, Ising evolution, Ising ground states, and 10-qubit for Heisenberg ground states, which are the same as the system sizes for training our method in the main text. Here we show additional results of finite measurements for 10-qubit systems as comparison to 50-qubit systems for Ising evolution, and Ising ground states. We do not include Heisenberg ground states because they are highly degenerate and can result in completely different ground state patterns for different system sizes, which prevents studying the affects of system sizes on measurement shots. Meanwhile, we do not include the IQP systems because they already work very well on 50-qubit system. The global fidelity with respect to the measurement shots for Ising evolution experiment are shown in the following table:
>
>
>
> | #measurement shots | 10-qubit systems | 50-qubit systems |
> | ------------------ | ---------------- | ---------------- |
> | **128**            | 0.7624 ± 0.2009  | 0.6783 ± 0.4439  |
> | **256**            | 0.8594 ± 0.2503  | 0.7198 ± 0.4006  |
> | **512**            | 0.9849 ± 0.0298  | 0.9754 ± 0.0098  |
> | **1024**           | 0.9903 ± 0.0115  | 0.9794 ± 0.0105  |
>
> For Ising ground states, the results are:
>
> | #measurement shots | 10-qubit systems | 50-qubit systems |
> | ------------------ | ---------------- | ---------------- |
> | **128**            | 0.9568 ± 0.0286  | 0.3565 ± 0.1107  |
> | **256**            | 0.9625 ± 0.0074  | 0.4505 ± 0.2720  |
> | **512**            | 0.9601 ± 0.0064  | 0.9193 ± 0.1373  |
> | **1024**           | 0.9578 ± 0.0060  | 0.9629 ± 0.0096  |
>
> From these results, we find that with the increase of system size, the required number of measurement shots for each observable increases for accurately reconstructing the target states.
>
> **5. For NISQ devices, how is the action space configured? Is it feasible to incorporate quantum channels into the action space?**
> **A5.** For NISQ devices, our method has an inherent purification property, which takes noisy data and produces unitary representations. While we could, in principle, use quantum channels as actions, the motivation for intentionally preserving the noise is unclear.
>
> **6. Previous work has leveraged prior knowledge, such as geometric locality (which the states examined in this paper also possess), to enhance the learning of quantum states and system properties. What advantages does the proposed method offer compared to these approaches?**
> **A6.** Thank you for the insightful question. As you mentioned, some approaches for learning quantum systems leverage geometric locality to enhance performance. While our model also incorporates prior knowledge in the design of the action space, the key advantage of our method is that it not only achieves satisfactory predictions of the properties but also explicitly constructs the circuit representations for the quantum states. The learned circuit can more easily be transferred to predict and learn other properties.

---

> ### Comment · Reviewer_m5ku · 2024-11-26
>
> Thank you to the authors for their explanations and additional experiments. While they have addressed some of my concerns, I still have the following question:
> 1. The states considered in the experiment in **A3** remain low-range entangled, which can be efficiently simulated using Matrix Product States (MPS). As a result, the necessity of learning these MPS representations based on measurement data remains questionable. It would be more convincing to conduct experiments on more general quantum states of larger size, such as those prepared using a random hardware-efficient ansatz.

---

### Official Review · Reviewer_89hB · 2024-11-04

**Soundness:** 2
**Presentation:** 1
**Contribution:** 2
**Rating:** 3
**Confidence:** 5

**Summary:**

In this paper, the authors propose a reinforcement learning (RL) based adaptive circuit construction approach for quantum measurement related tasks such as state or process tomography.
They aim to reconstruct any $n$-qubit pure state $|\psi\rangle$ by applying a unitary transformation $U$ to the computational basis state $|0\rangle^{\otimes n}$ and find the inverse unitary transformation.
Their method utilizes a reward function based on local fidelity and relies solely on two-body observables, which they claim results in polynomial sample complexity and avoids barren plateaus during optimization.
Numerical experiments of the proposed method are provided for learning of tasks upto $100$ qubits, specifically for states generated by Instantaneous Quantum Polynomial (IQP) circuits and time-evolved by many-body ground states of Ising and Heisenberg Hamiltonians.

**Strengths:**

- The integration of RL and ML for adaptive circuit design in quantum state tomography represents a novel contribution to the field. In short, the method is a demonstration of AI for Science as it applies RL to solving a quantum problem.

- The authors develop an explicit circuit representation that facilitates the reconstruction of target states and the computation of relevant properties. This foundation enhances the applicability of their method to downstream tasks, such as Hamiltonian learning, thereby extending its utility in quantum computing.

**Weaknesses:**

- The tomography task modeled within the RL framework is not clearly defined.
- Claims regarding polynomial sample complexity for learning an unknown state appear to contradict established lower bounds on sample complexity for full state tomography.
- The theoretical foundation (Proposition 1) suggesting that local fidelity can serve as a proxy for global fidelity has notable limitations:
  - This proposition holds primarily for area-law entangled states that are classically matrix product state (MPS) simulatable and may not apply to volume-law entangled states. As entanglement increases, local and global fidelity measures diverge, a limitation that should be explicitly discussed in the text.
  - Proposition 1 becomes vacuous for systems with large number of qubits, as it can be alluded from from Fig. 2. In the 4-qubit experiments, both local and global fidelities increase simultaneously. However, in the 50-qubit experiments, while local fidelity shows a consistent, monotonic increase, the global fidelity exhibits a sharp rise only at the end of the learning stage. The authors suggest that this abrupt increase in global fidelity is due to the presence of barren plateaus. This reasoning is somewhat unclear to me, as their initial motivation for minimizing local fidelity was based on the absence of barren plateaus.
  - If Proposition 1 is true then, it casts doubt on the claim of polynomial sample complexity. No theoretical or numerical evidence supporting this claim has been provided in the article.
- The authors have do not include most important baselines which are considered SOTA for the quantum state tomography problem: MPS QST and NNQS QSTs: restricted Boltzmann machines (RBM) and feed-forward neural networks with autoregressive architecture (ARN). Please have a look for relevant references within [1].
- Lack of open source code and reproducibility statement.

[1] Kurmapu, Murali K., et al. "Reconstructing complex states of a 20-qubit quantum simulator." PRX Quantum 4.4 (2023)

**Questions:**

- **Clarifications regarding the task**:
  - How are quantum circuits represented as RL states? What method does the RL agent use to select an action from this representation? For example, [2] employs one-hot encoding and tensor-based 3D encoding to represent circuits for neural network input. If such encodings are used then the authors should cite them as I think it is closely related to their work.
  - What is the specific metric being minimized, and to what level of precision is this metric evaluated for terminating the RL episode?
  - How are the parameters of the intermediate quantum circuits tuned or optimized during training? Are these parameters re-tuned when addressing downstream tasks?
  - Why is the action space (quantum gate set) customized for each task in your experiments? Can the authors present evidence that their method performs equivalently well when using a universal gate set $({RX(\theta), RY(\theta), RZ(\theta), CNOT})$ across all tasks?
  - The transformer-based measurement aggregator, which uses 2-local, nearest-neighbor Pauli measurements, lacks explanation. How do the authors derive local fidelity from these measurements? Do they estimate the average of local $|0\rangle\langle0|_i$ projections $Tr(O\rho)$ from the 2-local measurements? More detailed elaboration is required.

- Can the authors provide numerical or theoretical evidence supporting their claims regarding polynomial sample complexity in learning? At what stage do the authors start counting measurement samples—after the agent has been trained for a specific problem?

- Can the authors conduct experiments on state reconstruction within the QST problem for states that do not allow efficient MPS simulations (e.g., states with volume-law entanglement)?

- Can the authors provide numerical evidence to validate their claim of the existence of the barren plateaus in Fig. 2(b)?

- Why did the authors choose to compare their method only with SOTA baselines for the QST problem? Can they include comparisons with the methods highlighted in the Weaknesses section?

[2] Patel, Yash J., et al. "Curriculum reinforcement learning for quantum architecture search under hardware errors" ICLR (2024)

---

> ### Author Response · Authors · 2024-11-25
> **Response to Reviewer 89hB (Part 1)**
>
> We deeply appreciate the reviewer’s valuable feedback and careful review; we are addressing the issues systematically and have incorporated the necessary changes in the revised PDF, with all updates highlighted in blue for ease of reference.
>
> **Weaknesses:**
>
> 1 and 2. **The tomography task modeled within the RL framework is not clearly defined.
> Claims regarding polynomial sample complexity for learning an unknown state appear to contradict established lower bounds on sample complexity for full state tomography.**
>
> **A1 and A2.** Thank you for the question. It is important to note that we are not performing full state tomography. Our motivation is to learn how to locally reconstruct the target family of states using quantum circuits, where specific local properties can be estimated accurately. The learning process relies solely on local observables for the target states of interest. While the sample complexity for full state tomography is $\mathcal{O}(2^N)$, if we are only concerned with local properties, the complexity reduces to $\mathcal{O}(\text{poly}(N))$ or even $\mathcal{O}(N)$ (with geometric constraints). This applies when specific properties of the states under certain local observables are of interest. In such cases, full state tomography is unnecessary; instead, we can focus on constructing the reduced density matrices for the subsystems of interest.
>
> 3. **The theoretical foundation (Proposition 1) suggesting that local fidelity can serve as a proxy for global fidelity has notable limitations:
>    This proposition holds primarily for area-law entangled states that are classically matrix product state (MPS) simulatable and may not apply to volume-law entangled states. As entanglement increases, local and global fidelity measures diverge, a limitation that should be explicitly discussed in the text.**
>
> **A3.1** We agree that local and global fidelity diverges as the entanglement increases, thus our method may not be applied to tomographically reconstruct highly entangled states, but it still works for predicting local properties of interest. As for Proposition 1, note that $1 - N\epsilon$ is a lower bound for global fidelity, and the derivation does not depend on the assumption of the entanglement of the target states. One would expect this being the worst-case performance, namely when the state is very entangled. If the model learns a good approximation of the local fidelity, the global fidelity only diverges linearly with the system size.
>
> **Proposition 1 becomes vacuous for systems with a large number of qubits, as it can be alluded from Fig. 2. In the 4-qubit experiments, both local and global fidelities increase simultaneously. However, in the 50-qubit experiments, while local fidelity shows a consistent, monotonic increase, the global fidelity exhibits a sharp rise only at the end of the learning stage. The authors suggest that this abrupt increase in global fidelity is due to the presence of barren plateaus. This reasoning is somewhat unclear to me, as their initial motivation for minimizing local fidelity was based on the absence of barren plateaus.**
>
> **A3.2** The motivation for optimizing local fidelity is that "when local fidelity is high, one would expect global fidelity to be not too low." The applicable condition for Proposition 1 is when the model has learned a good approximation for the local fidelity. For large-scale systems where barren plateaus exist (like the system in Fig. 2(b)), optimizing global fidelity is hard since it is difficult for the agent to gain higher reward using global fidelity during exploration because most of the states have nearly zero fidelity with the target state unless the state is sufficiently close to the target. Therefore, we choose to optimize local fidelity because as the current state approaches the target state, the local fidelity monotonically increases. Fig. 2(b) shows that when the local fidelity becomes high enough during training, the global fidelity also increments. This result is coherent with Proposition 1, which states that if the agent learns a good approximation of the local fidelity, the global fidelity can be bounded.
>
> **If Proposition 1 is true, then it casts doubt on the claim of polynomial sample complexity. No theoretical or numerical evidence supporting this claim has been provided in the article.**
>
> **A3.3** We count the number of observables we use for sample complexity. In our settings, we only consider local Pauli observables as described in Section 2.2. The number of observables scales linearly with the system size. However, we realize that using the term "sample complexity" is not rigorous enough in our settings because many theorists care about the total number of states required in the learning procedure, but it is hard to give a rigorous bound on this resource under the settings of reinforcement learning. Therefore, we change "sample complexity" into "number of observables."

---

> ### Author Response · Authors · 2024-11-25
> **Response to Reviewer 89hB (Part 2)**
>
> 4. **The authors do not include the most important baselines which are considered SOTA for the quantum state tomography problem: MPS QST and NNQS QSTs: restricted Boltzmann machines (RBM) and feed-forward neural networks with autoregressive architecture (ARN). Please have a look for relevant references within [1].**
>
> **A4.** The TQS (Transformer Quantum State) used in our comparisons is a specific type of NNQST that employs a Transformer as the neural network architecture. It learns the states using the same training strategy as NNQST, which involves Monte Carlo estimation of the objective function—in our case, the global fidelity. This approach is essentially similar to ARN or RBM, where a neural network model is used to represent the probability amplitude of quantum states. For MPS QST, our simulation is based on representing quantum states using MPS and applying MPO to MPS in the same way that gates are applied to quantum states. In this sense, our method implicitly incorporates MPS QST and can further decode the circuit that prepares the MPS.
>
> 5. **Lack of open source code and reproducibility statement.**
>
> **A5.** This is mainly due to the double-blind requirement. We will release the source code upon acceptance of this paper or when the requirement no longer applies.
>
> **Questions:**
>
> 1. **Clarifications regarding the task:
>    How are quantum circuits represented as RL states? What method does the RL agent use to select an action from this representation? For example, [2] employs one-hot encoding and tensor-based 3D encoding to represent circuits for neural network input. If such encodings are used then the authors should cite them as I think it is closely related to their work.**
>
> **A1.1** Thanks for the questions. In our approach, we do not use quantum circuits as RL states. We do not use the term "state" but "observation" to refer to the input of the agent. In RL terminology, "observation" contains partial information of the "state." The "state" in our framework refers to the quantum state, and "observation" corresponds to the measurement data obtained by performing local observables to the quantum state. Since using quantum states directly requires tomography, which is not practical in general, we hide the "state" into the environment, allowing the agent to perceive only partial information as observation. The "state" is represented by MPS in our simulation, but we highlight that in a real-world scenario, this could also be an unknown quantum state generated by a quantum circuit. After obtaining the observation, the agent processes the values as a sequence of tokens, similar to how natural languages are processed, and selects an action from a pre-defined dictionary that maps the neural network's output to a quantum gate type along with the corresponding parameters. This step does not require additional encoding methods such as one-hot or tensor-based 3D encoding in [2], but rather resembles the behavior of large language models, where the input and output are sequences.
>
> **What is the specific metric being minimized, and to what level of precision is this metric evaluated for terminating the RL episode?**
>
> **A1.2** The metric is the local fidelity reward function defined in Equation (7). The agent is trained to maximize this reward. The RL episode terminates when the average local fidelity (Equation 4) exceeds a threshold of 0.999, or when the episode length exceeds $T$ as stated in the Experiment section.
>
> **How are the parameters of the intermediate quantum circuits tuned or optimized during training? Are these parameters re-tuned when addressing downstream tasks?**
>
> **A1.3** As stated in our framework, the parameters of the quantum circuits are updated by the RL agent. The agent can be viewed as a meta-learner, which learns how to adjust these parameters based on the local fidelity reward. No gradient calculations are required for the parameters. After training, the parameters are fixed and used for predicting properties and addressing downstream tasks.

---

> ### Author Response · Authors · 2024-11-25
> **Response to Reviewer 89hB (Part 3)**
>
> **Why is the action space (quantum gate set) customized for each task in your experiments? Can the authors present evidence that their method performs equivalently well when using a universal gate set across all tasks?**
>
> **A1.4** In fact, our proposed method does not require the action space to be designed to match the preparation circuits. We conducted an experiment using a universal set of one- and two-qubit gates formed by $g=\exp(i\theta G)$ and $G = \{X, Y, Z\} \cup \{X,Y,Z\}^{\otimes 2}$, where $\theta \in [-\pi / 2, \pi / 2]$. The state family we consider is a mixture of Ising and Heisenberg ground states of 4 qubits. The coefficients of Hamiltonians are the same as those in the Experiment section of the main text. The results are shown in the table below:
>
> | Experiment     | System size | Fidelity        | Rényi Entropy | Two-point Correlations | Spin-Z |
> | -------------- | ----------- | --------------- | ------------- | ---------------------- | ------ |
> | Mixture family | 4           | 0.9587 ± 0.0130 | 0.0745        | 0.0128                 | 0.0434 |
>
> We also note that in many practical scenarios, some prior information is available to inform the choice of action space. For example, it is often possible to learn the ground states of a Heisenberg-interaction many-body system without knowing the interaction coefficients but knowing the skeleton of the Hamiltonian.
>
> **The transformer-based measurement aggregator, which uses 2-local, nearest-neighbor Pauli measurements, lacks explanation. How do the authors derive local fidelity from these measurements? Do they estimate the average of local projections from the 2-local measurements? More detailed elaboration is required.**
>
> **A1.5** The decoding of actions from measurement data using the transformer-based neural network and the computing of local fidelity are actually two distinct procedures. The 2-local measurement data is used solely as input to the agent. The average local fidelity is obtained by measuring the state with the observables defined in Equation 5 and Equation 6. In our setting, the number of observables scales linearly with the system size. After applying the observable $|0\rangle \langle 0|$ to each qubit of the state, the average value is taken as the average local fidelity.
>
> 2. **Can the authors provide numerical or theoretical evidence supporting their claims regarding polynomial sample complexity in learning? At what stage do the authors start counting measurement samples—after the agent has been trained for a specific problem?**
>
> **A2.** We do not focus on the sample complexity for the learning process, but rather on the measurement aspect. It is difficult to provide a rigorous bound for the RL learning process. We count the measurement samples as the number of observables required to measure the states. We acknowledge that the term "sample complexity" is not precise in our setting, and as such, we have replaced it with "number of observables" for clarity.
>
> 3. **Can the authors conduct experiments on state reconstruction within the QST problem for states that do not allow efficient MPS simulations (e.g., states with volume-law entanglement)?**
>
> **A3.** The current simulation backbone of our framework relies on MPS to demonstrate performance on large-scale systems. While our proposed framework can naturally be extended to real quantum states, as the interface (both the input to the agent and the output) can be directly applied to them, conducting such experiments is currently not feasible due to time limitations and the lack of experimental data. We would improve this in future work by conducting experiments on real quantum computers. Thank you for your understanding.
>
> 4. **Can the authors provide numerical evidence to validate their claim of the existence of the barren plateaus in Fig. 2(b)?**
>
> **A4.** We appreciate the reviewer's question. Fig. 2(b) itself is evidence of the existence of barren plateaus. Note that barren plateaus refer to a large region of nearly zero gradient of the cost function. In Fig. 2(b), the x-axis is the iterative steps, each of which represents a specific state. This can be viewed as a trajectory from the initial state $|\psi\rangle$ to the target state $|0\rangle$. From step 0 to step 60, regardless of how the circuit evolves, the global fidelity remains close to 0. This suggests that the landscape of the global fidelity objective function is flat, with values predominantly at 0, resulting in a near-zero gradient over a large region. It is only towards the end of the episode, when the reconstructed state is sufficiently close to the target, that the objective function experiences a sharp increase.

---

> ### Author Response · Authors · 2024-11-25
> **Response to Reviewer 89hB (Part 4)**
>
> 5. **Why did the authors choose to compare their method only with SOTA baselines for the QST problem? Can they include comparisons with the methods highlighted in the Weaknesses section?**
>
> **A5.** We have not mentioned any specific method for the QST problem in the weaknesses section, but only some directions or techniques that could be involved in our framework for improvement. As mentioned earlier, our framework does not perform full state tomography.

---

> > ### Comment · Reviewer_89hB · 2024-11-29
> > **Response to the authors**
> >
> > We appreciate the authors’ efforts to address the concerns raised in the initial review. Their responses have clarified some aspects of the paper, and I now have a better understanding of their approach. However, I remain unable to raise my initial evaluation due to fundamental issues that persist. Below, I outline these concerns in detail.
> >
> > ### **Polynomial Sample Complexity**
> > The authors’ revised claim that only $O(\text{poly}(N))$ or $O(N)$ local observables are required—rather than sample complexity—does not guarantee that the output is always approximately correct or within an acceptable error $\epsilon$. As Reviewer WRZ5 also noted, the choice of $O(\text{poly}(N))$ observables appears to be by *construction*, which does not necessarily ensure that the result is always reliable. In contrast, methods specifically designed for such tasks, such as classical shadows, come with guarantees. While the authors present their method as a heuristic, one would expect a desirable property for heuristics to be that increasing resources improves the quality of the solution. This guarantee is absent in the current approach.
> >
> > ### **Proposition 1**
> > The use of local fidelity as a proxy for global fidelity introduces significant limitations. The authors acknowledge that this technical assumption breaks down as entanglement increases. For instance, consider the 4-qubit GHZ state $| \psi \rangle = |0000\rangle + |1111\rangle$ and another state $| \psi' \rangle = |0101\rangle + \exp(-i\theta)|1010\rangle$. These states have the same local fidelity, but their global fidelities differ, with both being far from $|0\rangle^{\otimes 4}$. This mismatch suggests that the method's reliance on local fidelity may lead to significant inaccuracies in certain cases.
> >
> > ### **Relevance of Reconstructed States and Baselines**
> > The proposed method is limited to reconstructing quantum states realizable by matrix product states (MPS). However, the authors compare their approach to methods that are not restricted to MPS-representable states, potentially skewing the comparison. For a fairer evaluation, the authors could compare their method to approaches explicitly developed for MPS-realizable states. Alternatively, as a future direction, the authors might include numerical simulations involving states with volume-law entanglement to demonstrate the broader applicability of their method. This would better contextualize the utility of their approach relative to the state of the art.
> >
> > ### **Claims Regarding Figure 2 as Evidence of Barren Plateaus**
> > To my knowledge, there is no established connection between local fidelity and the variance of gradients, which is the key quantity used to define barren plateaus. If the authors claim that Figure 2(b) provides evidence of barren plateaus, they should directly demonstrate this by plotting the variance of the gradients of the circuit. Without such evidence, the claim remains speculative and unsupported.

---

### Official Review · Reviewer_TK92 · 2024-11-07

**Soundness:** 2
**Presentation:** 3
**Contribution:** 2
**Rating:** 6
**Confidence:** 5

**Summary:**

The work focusses on learning circuits for characterizing quantum states. Explicit representation of quantum
states using circuits are provided. RL is used with a local fidelity reward function.Experiments are performed
for quantum states up to 100 qubits.Results on ground state and Hamiltonian learning are presented.

**Strengths:**

1. Local fidelity reward function proposed.
2. A theoretical guarantee for the learning function provided.
3. The ground states of the Ising and Heisenberg XYZ ground states are learnt well.

**Weaknesses:**

1. Results are presented for shallow depth circuits. No indication is provided
on the increase of complexity with depth.
2. For the Hamiltonian learning, the circuits parameters are failed to be learnt from the
linear model using VQE and QAOA. This puts some limitation on applicability.

**Questions:**

1. Would some nonlinear model work for VQE and QAOA?

---

> ### Author Response · Authors · 2024-11-25
> **Response to Reviewer TK92**
>
> We sincerely thank the reviewer for their thoughtful feedback and detailed review; we are addressing the issues step by step, and have made the necessary changes in the revised PDF, highlighting them in blue for clarity.
>
> **Weaknesses:**
>
> 1. **Results are presented for shallow depth circuits. No indication is provided on the increase of complexity with depth.**
>
> **A1.** Our method requires measuring the state after one application of a column of gates to the state. Since we consider two-local measurements, the total number of observables used to measure the state is `9(N-1)`, where `N` is the number of qubits and `9` refers to 9 different combinations of Pauli X, Y, and Z observables. Suppose the circuit has `T` layers, then the total number of observables required for evolving the initial state to the target state is `9T(N-1)`. This indicates the measurement complexity scales linearly with the circuit depth.
>
> 2. **For the Hamiltonian learning, the circuits parameters are failed to be learnt from the linear model using VQE and QAOA. This puts some limitation on applicability.**
>
> **A2.** Thanks for raising this question.
> There might be a misunderstanding on what exactly the downstream task is. Here the task is to learn the Hamiltonian coefficients, given the circuit representations of the ground states.  Results in **Figure 6** show the performance of our method (Circuit Rep) is much better in comparison with combining existing methods of circuit representation (QAOA and VQE) with a linear regression model. This shows that the circuit parameters generated by our method can be successfully utilized to predict the Hamiltonian coefficients, demonstrating the applicability of our method instead of putting some limitation on it.
>
> Meanwhile, we highlight that **Figure 6** only shows the test performance on using circuit parameters for Hamiltonian learning. In fact, for training, all methods achieve high precision on predicting the Hamiltonian coefficients using a linear model. The following table shows the mean squared error of predicting Hamiltonian coefficients on the training set:
>
> | System      | Ours     | VQE       | QAOA      |
> |-------------|----------|-----------|-----------|
> | **Ising**   | 1.758e-14 | 8.382e-32 | 1.208e-28 |
> | **Heisenberg** | 3.200e-14 | 1.887e-12 | 7.889e-32 |
>
> This indicates that one can utilize the circuit parameters to conduct Hamiltonian learning. However, the QAOA and VQE parameters fail to generalize to the test cases.
>
> ---
>
> **Questions:**
>
> 1. **Would some nonlinear model work for VQE and QAOA?**
>
> **A.** We have also done experiments using more complex models like polynomial regression and MLP, but found that the linear model is the best for QAOA and VQE.

---

### Meta-Review · Area_Chair_BC3r · 2024-12-19

**Metareview:**

The submission proposes a reinforcement learning (RL) framework for constructing explicit circuit representations of quantum states using local fidelity as a reward function. While the integration of RL and quantum state learning is innovative, the paper faced concerns from reviewers, including weak theoretical support for claims about polynomial sample complexity and barren plateau mitigation, the limited scope of numerical experiments (focused on MPS-representable states), and the lack of comparisons with key baselines.

Despite revisions and additional clarifications during the rebuttal process, these fundamental issues remain unresolved for most reviewers, leading to mixed evaluations. The numerical results demonstrate promise, especially for systems with up to 100 qubits, but their relevance is questioned due to the tailored setup and absence of guarantees for general applicability. The lack of open-source code and reproducibility further diminishes the work's impact.

**Additional Comments On Reviewer Discussion:**

During the rebuttal period, the authors addressed multiple points raised by the reviewers and provided additional experiments to support their claims. However, several concerns remain unresolved.

**Claims on polynomial sample complexity and barren plateau mitigation (raised by 89hB, WRZ5, VHHj)**

Reviewers expressed concerns about the theoretical support for these claims and noted contradictions with known bounds. The authors clarified that they do not perform full-state tomography and that their method focuses on local fidelity and the number of observables, not overall sample complexity. They also rephrased "circumvent barren plateaus" to "mitigate barren plateaus." While the clarifications provided additional context, the responses did not fully resolve the reviewers' concerns.

**Limited scope of simulations (raised by 89hB, WRZ5, m5ku)**

Reviewers criticized the focus on MPS-representable states, which limits generalizability. The authors acknowledged this limitation and provided explanations on how the method could be extended in future work. However, no experiments on non-MPS states were conducted due to time constraints. The tailored setup and narrow experiments' scope reduced the results' perceived impact and applicability.

---

### Decision · Program_Chairs · 2025-01-22

Reject